# UBXN3B positively regulates STING-mediated antiviral immune responses

Long Yang[1,2], Leilei Wang[1,3], Harshada Ketkar[1], Jinzhu Ma[1,4], Guang Yang[5], Shuang Cui[6], Tingting Geng [1], Dana G. Mordue[1], Toyoshi Fujimoto[7], Gong Cheng[8], Fuping You[6], Rongtuan Lin[2], Erol Fikrig[9,10] & Penghua Wang[1]

The ubiquitin regulatory X domain-containing proteins (UBXNs) are likely involved in diverse biological processes. Their physiological functions, however, remain largely unknown. Here we present physiological evidence that UBXN3B positively regulates stimulator-of-interferon genes (STING) signaling. We employ a tamoxifen-inducible Cre-LoxP approach to generate systemic *Ubxn3b* knockout in adult mice as the *Ubxn3b*-null mutation is embryonically lethal. *Ubxn3b$^{-/-}$*, like *Sting$^{-/-}$* mice, are highly susceptible to lethal herpes simplex virus 1 (HSV-1) and vesicular stomatitis virus (VSV) infection, which is correlated with deficient immune responses when compared to *Ubxn3b$^{+/+}$* littermates. HSV-1 and STING agonist-induced immune responses are also reduced in several mouse and human *Ubxn3b$^{-/-}$* primary cells. Mechanistic studies demonstrate that UBXN3B interacts with both STING and its E3 ligase TRIM56, and facilitates STING ubiquitination, dimerization, trafficking, and consequent recruitment and phosphorylation of TBK1. These results provide physiological evidence that links the UBXN family with antiviral immune responses.

[1] Department of Microbiology and Immunology, New York Medical College, 15 Dana Road, Valhalla, NY 10595, USA. [2] Lady Davis Institute-Jewish General Hospital, Department of Medicine, McGill University, 3755 Chemin de la Côte-Sainte-Catherine, Montreal, QC H3T 1E2, Canada. [3] Department of Obstetrics and Gynecology, Shengjing Hospital, China Medical University, 110004 Shenyang City, Liaoning Province, China. [4] College of Life Science and Technology, Heilongjiang Bayi Agricultural University, 163319 Daqing City, Heilongjiang Province, China. [5] Department of Parasitology, School of Medicine, Jinan University, 510610 Guangzhou City, Guangdong Province, China. [6] Beijing Key Laboratory of Tumor Systems Biology, Department of Immunology, School of Basic Medical Sciences, Institute of Systems Biomedicine, Peking University Health Science Center, 100083 Beijing, China. [7] Department of Anatomy and Molecular Cell Biology, Nagoya University Graduate School of Medicine, 65 Tsurumai, Showa, Nagoya 466-8550, Japan. [8] Tsinghua-Peking Center for Life Sciences, School of Medicine, Tsinghua University, 100084 Beijing, China. [9] Section of Infectious Diseases, Yale University School of Medicine, 333 Cedar Street, New Haven, CT 06510, USA. [10] Howard Hughes Medical Institute, 4000 Jones Bridge Road, Chevy Chase, MD 20815, USA. Correspondence and requests for materials should be addressed to L.Y. (email: longyangrich@hotmail.com) or to E.F. (email: erol.fikrig@yale.edu) or to P.W. (email: Penghua_wang@nymc.edu)

The mammalian innate immune system detects invading pathogens via several families of pattern recognition receptors, one of which the cyclic GMP-AMP synthase senses cytosolic DNA[1] and synthesizes cGAMP[2,3]. cGAMP serves as a second messenger which binds an endoplasmic reticulum (ER) membrane adaptor, stimulator-of-interferon genes (STING), to trigger antiviral type I interferon (IFN-I) and inflammatory immune responses[4–6]. STING undergoes lysine K63-linked polyubiquitination mediated by E3 ligase tripartite motif containing 32 (TRIM32) and TRIM56[7,8], dimerizes[7], and traffics out of the ER to perinuclear vesicles[6,9,10] where it recruits and activates the kinase TANK-binding kinase 1 (TBK1) and inhibitor of nuclear factor κB kinase[5,11], and in turn innate immune responses. STING could also transmit signals from other DNA sensors such as DNA-dependent activator of IFN regulatory factors (IRFs), DEAD-Box helicase 41, DNA-dependent protein kinase, and IFN-γ-inducible protein 16; however, their function in IFN-I response is still controversial[12]. Being such a central adaptor in the DNA sensing pathway, STING signaling is stringently regulated by post-translational modifications. TBK1-mediated phosphorylation of STING provides a "licensing" mechanism for TBK1 to phosphorylate IRF3 and induction of IFN-I[13], while nucleotide-binding oligomerization domain receptor X1 sequesters STING to negatively regulate its signaling[14]. K63-linked polyubiquitination by TRIM32, TRIM56, and mitochondrial E3 ubiquitin protein ligase 1 (MUL1), K27 polyubiquitination by ER-resident E3 ligase complex autocrine motility factor receptor (AMFR)–insulin-induced gene 1 (INSIG1) promotes STING signaling[7,8,15], while K48-linked polyubiquitination of STING by E3 ligases Ring finger protein 5 and TRIM30α dampens its signaling by targeting it for proteasomal degradation[16,17]. Obviously, regulation of STING by ubiquitination is very complex and the in vivo physiological relevance of different types of ubiquitination and E3 ligases remains elusive.

The ubiquitin regulatory X (UBX) domain shares weak homology with ubiquitin at the protein level and adopts the same three-dimensional-fold as ubiquitin[18]. One of the best understood UBXN proteins, p47, is a cofactor of the AAA ATPase (ATPase associated with various cellular activities), p97[19]. p97 is highly conserved across species and involved in diverse cellular processes, including ubiquitin-dependent protein degradation, vesicle fusion, and cell cycle[18]. Several UBXN proteins have been recently shown to bind p97 and a large number of E3 ubiquitin ligases[20], suggesting a potential role for UBXNs in regulation of global protein turnover and cellular signaling. However, the molecular function of the majority of UBXN proteins is largely unknown. Here we perform a family-based screening to identify a UBXN that could modulate STING-dependent signaling. Indeed, by utilizing dual luciferase reporter system, we identify that UBXN3B is a positive regulator of STING and that UBXN3B complexes with STING and TRIM56 to potentiate STING-dependent innate immune responses.

## Results

**UBXN3B regulates STING-dependent IFN-I induction.** We performed an IFN-stimulated response element (ISRE) reporter activation screening in human embryonic kidney 293 cells transformed with T antigen of SV40 (HEK293T) cells for UBXNs that enhanced a STING-mediated IFN response. Transfection with HA-STING plasmid and vector induced ISRE activity by ~150-fold relative to vector alone (Supplementary Fig. 1a). Interestingly, among all the UBXNs tested, UBXN3B most significantly potentiated a STING-induced IFN-I response, which was ~5-fold higher than STING alone (Supplementary Fig. 1a).

Overexpression of UBXN3B alone did not induce ISRE significantly (Supplementary Fig. 1b–d), indicating that UBXN3B is a positive regulator of STING-dependent signaling. To validate the screening results, we used cGAMP, a second messenger that binds and stimulates STING signaling[1,21]. cGAMP was used to prime HEK293T cells stably expressing FLAG-STING (designated HEK293T-STING cell line) following transfection with a UBXN3B expression plasmid. We found that UBXN3B synergized with cGAMP to induce an IFN-I response in a dose-dependent manner (Supplementary Fig. 1b and 1c). To confirm that the enhancing effect on STING signaling was specific to UBXN3B, we included two UBXN members, UBXN3A which is the closest sibling of UBXN3B and UBXN9 which is more dissimilar from UBXN3B, in terms of domain structure[20]. Indeed, only UBXN3B, but not UBXN3A or UBXN9, enhanced the cGAMP-induced IFN-I response (Supplementary Fig. 1d). In addition, we tested if UBXN3B has a modulatory effect on other important antiviral signaling pathways. The results demonstrate that UBXN3B failed to synergize with retinoic acid-inducible gene I (RIG-I)-like receptor (RLR)-mediated and Toll-like receptor (TLR)-mediated IFN-I induction (Supplementary Fig. 2). These data suggest that UBXN3B specifically targets STING signaling.

**UBXN3B is critical for anti-DNA virus immune responses.** UBXN3B, also known as Fas-associated factor 2 and UBXD8, might play a role in regulating lipid metabolism[22–24]. However, its physiological function remains largely unclear. We therefore attempted to generate general $Ubxn3b^{-/-}$ mice using the CRISPR-Cas9 technology and unfortunately failed, suggesting that Ubxn3b deletion is embryonically lethal, consistent with a recent publication[24]. We therefore planned to generate an inducible conditional knockout strain so as to systemically ablate UBXN3B gene expression. We obtained $Ubxn3b^{flox/flox}$ (exon 1, homologous recombination) from our collaborator[24] and crossed it with a tamoxifen-inducible Cre mouse line[25,26], to generate a $Cre^{+/-}Ubxn3b^{flox/flox}$ mouse. The tamoxifen-inducible Cre model has been successfully applied to many gene deletions including essential genes[25]. Indeed, Ubxn3b protein expression was abolished in various tissues of tamoxifen-treated Cre$^{+/-}Ubxn3b^{flox/flox}$ mice (Fig. 1a). However, no discernible developmental defects or behavioral abnormality were observed in these mice. We then infected tamoxifen-pretreated $Cre^{+/-}Ubxn3b^{flox/flox}$ (referred to as $Ubxn3b^{-/-}$ for convenience throughout the manuscript) or untreated $Cre^{+}Ubxn3b^{flox/flox}$ (referred to as $Ubxn3b^{+/+}$) littermates with a DNA virus, herpes simplex virus 1 (HSV-1), which effectively induces STING signaling. To exclude any potential side effects of tamoxifen on HSV-1 pathogenesis and/or antiviral immunity, we included tamoxifen-treated $Cre^{+/-}$ and $Ubxn3b^{flox/flox}$ littermates as controls. As shown in Fig.1b, $Ubxn3b^{-/-}$ mice ($N = 8$) died of HSV-1 infection earlier than $Ubxn3b^{+/+}$ mice ($N = 8$) and the other control mice ($N = 9$). The overall survival rate of $Ubxn3b^{-/-}$ mice was also much lower ($P < 0.01$, log-rank test). In addition, all infected $Ubxn3b^{-/-}$ mice developed severe neurological symptoms, as well as hunched posture, decreased movement, and labored breathing on day 3 after infection. These symptoms were completely absent from $Ubxn3b^{+/+}$ animals (Supplementary Movie 1). Compared to $Sting^{-/-}$ mice ($N = 8$), $Ubxn3b^{-/-}$ mice ($N = 9$) were a bit less sensitive to HSV-1 infection (Fig. 1c). In agreement with the survival data, the serum IFN-I abundance was markedly decreased at 4 and 8 h after HSV-1 inoculation in $Ubxn3b^{-/-}$ ($N = 6$) compared to $Ubxn3b^{+/+}$ ($N = 10$) mice (non-parametric Mann–Whitney test, $P < 0.05$), demonstrating that UBXN3B is critical for HSV-1-induced IFN response (Fig. 1d). In agreement with this, the mRNA levels of *Ifnb1*, *Tnfa*,

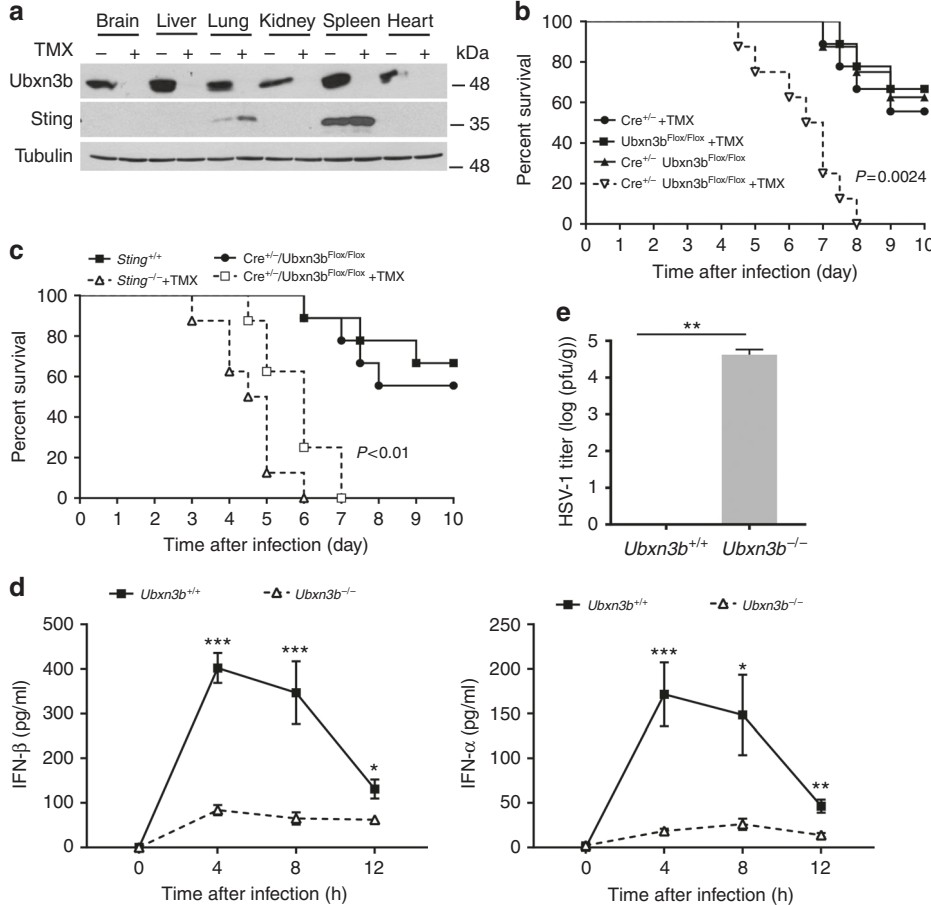

**Fig. 1** UBXN3B is critical for IFN-I induction by and control of HSV-1 infection in vivo. **a** Immunoblots showing *Ubxn3b* knockout efficiency in various tissues without/with tamoxifen (TMX) treatment in Cre[+/−] *Ubxn3b*[flox/flox] mice. Tubulin is a housekeeping protein control. **b, c** The survival curves of tamoxifen (TMX)-treated *Ubxn3b*[flox/flox], TMX-treated Cre[+/−], mock-treated Cre[+/−] *Ubxn3b*[flox/flox] (designated *Ubxn3b*[+/+]) and TMX-treated Cre[+/−] *Ubxn3b*[flox/flox] (designated *Ubxn3b*[−/−]) littermate and *Sting*[−/−] mice challenged with $1 \times 10^7$ plaque-forming units (PFU) per mouse of HSV-1 i.v. N = 8–9 mice/genotype. ***P < 0.001 (log-rank test). **d** The serum IFN-I concentrations (mean ± s.e.m) of mice challenged with $2 \times 10^6$ PFU per mouse of HSV-1 i.v. **P < 0.01; *P < 0.05 (non-parametric Mann–Whitney test), N = 10 (*Ubxn3b*[+/+]) or N = 6 (*Ubxn3b*[−/−]). **e** The viral titers (mean ± s.e.m) in the brain on day 3 after infection (PFU per gram tissue). **P < 0.01 (non-parametric Mann–Whitney test), N = 5 mice per genotype. The data are pooled from two independent experiments

and *Ifit1* (an IFN-I-induced antiviral effector) in leukocytes of *Ubxn3b*[−/−] mice were also lower than those in *Ubxn3b*[+/+] littermates (N = 5 each genotype, P < 0.05, non-parametric Mann–Whitney test) (Supplementary Fig. 3). Given that HSV-1 can penetrate the brain and elicit fatal encephalitis in mice, we then quantified HSV-1 titers in the brain by plaque-forming assay. *Ubxn3b*[−/−] mice had a significantly increased number of viral particles compared to *Ubxn3b*[+/+] mice on day 3 after infection (N = 5 each genotype, P < 0.01, non-parametric Mann–Whitney test) (Fig. 1e).

We next asked if the antiviral function of UBXN3B was cell-type specific. We isolated bone marrow cells from Cre[+/−] *Ubxn3b*[flox/flox] mice pretreated without (*Ubxn3b*[+/+]) or with tamoxifen (*Ubxn3b*[−/−]) and differentiated them into macrophages (bone marrow-derived macrophage (BMDMs)) with L929 cell-conditioned medium, conventional dendritic cells (cDCs) with granulocyte–macrophage colony-stimulating factor (GM-CSF) and plasmacytoid DC (pDCs) with Flt3L. Induction of IFN-I protein by HSV-1, STING-stimulating immunostimulatory DNA (ISD) and cGAMP was remarkably compromised in knockout cells (Fig. 2a). Correspondingly, induction of Oas1a protein, an IFN-stimulated gene (ISG), was also impaired in knockout cDC cells (Fig. 2b). The mRNA levels of *Ifnb1* and *Tnfa*

were decreased in *Ubxn3b*[−/−] cells at 6 h after infection, which was accompanied by increased viral load (Fig. 2c, d). The above-mentioned immune cells are not very permissive to HSV-1 infection. We next isolated mouse embryonic fibroblasts (MEFs), which are highly permissive to HSV-1 in contrast to the above-mentioned immune cells we induced in vitro, from untreated Cre[+/−] *Ubxn3b*[flox/flox] E14 embryos and induced *Ubxn3b* deletion with 4-hydroxyl tamoxifen in vitro. Ubxn3b protein expression was almost completely depleted after 4-hydroxyl tamoxifen treatment in Cre[+/−] *Ubxn3b*[flox/flox] (*Ubxn3b*[−/−]) cells (Fig. 2e). In agreement with the aforementioned results, the viral titers produced by *Ubxn3b*[−/−] cells were higher than *Ubxn3b*[+/+] (Fig. 2f). *Ifnb1* and its ISG mRNA levels induced by HSV-1 were consistently down-regulated in *Ubxn3b*[−/−] cells (Fig. 2g).

The UBXN3B protein is evolutionarily conserved, with 97% identity between human and rodent. We then asked if its antiviral function is also evolutionarily conserved. We employed the CRISPR-Cas9 technology to generate *UBXN3B*[−/−] in H1975 cell, a human lung epithelial cell line (Supplementary Fig. 4a). Consistent with the results from mouse cells, *IFNB1* induction by ISD was much lower in *UBXN3B*[−/−] cells than wild type (WT) cells. But the dsRNA analog polyinosinic:polycytidylic acid

(polyIC)-induced *IFNB1* expression was not significantly impaired (Supplementary Fig. 4b). The HSV-1 titers (Supplementary Fig. 4c) and intracellular HSV-1 protein levels (Supplementary Fig. 4d) were very much increased in *UBXN3B*[−/−] cells compared to WT cells, suggesting a critical antiviral role for

UBXN3B. The immune responses to HSV-1 including IFN-I (*IFNB1*) and inflammatory response (*TNFA*) were reduced in the absence of UBXN3B expression (Supplementary Fig. 4e). To extend this further, we next examined the UBXN3B antiviral function in primary human trophoblasts. These cells (1) may be

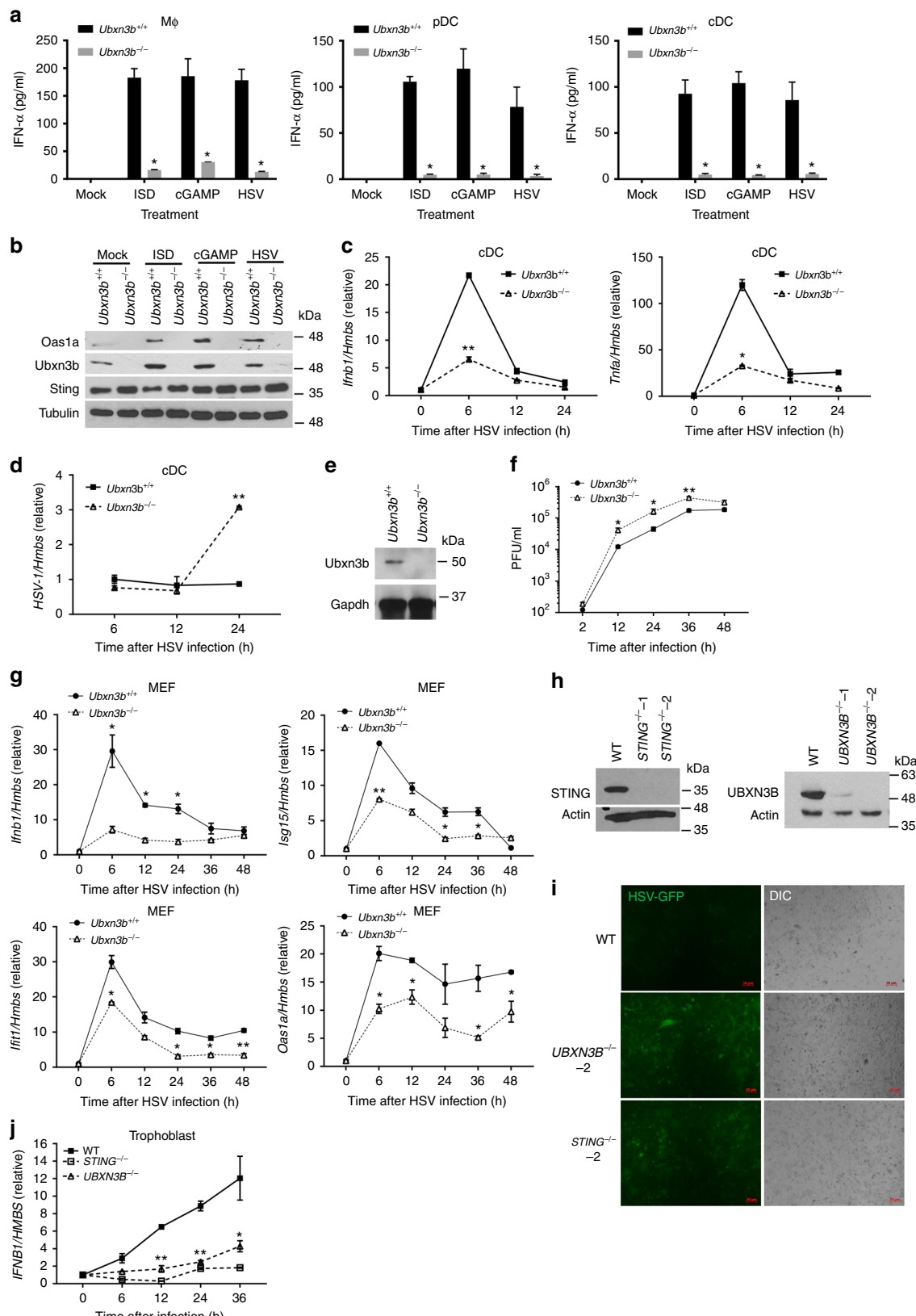

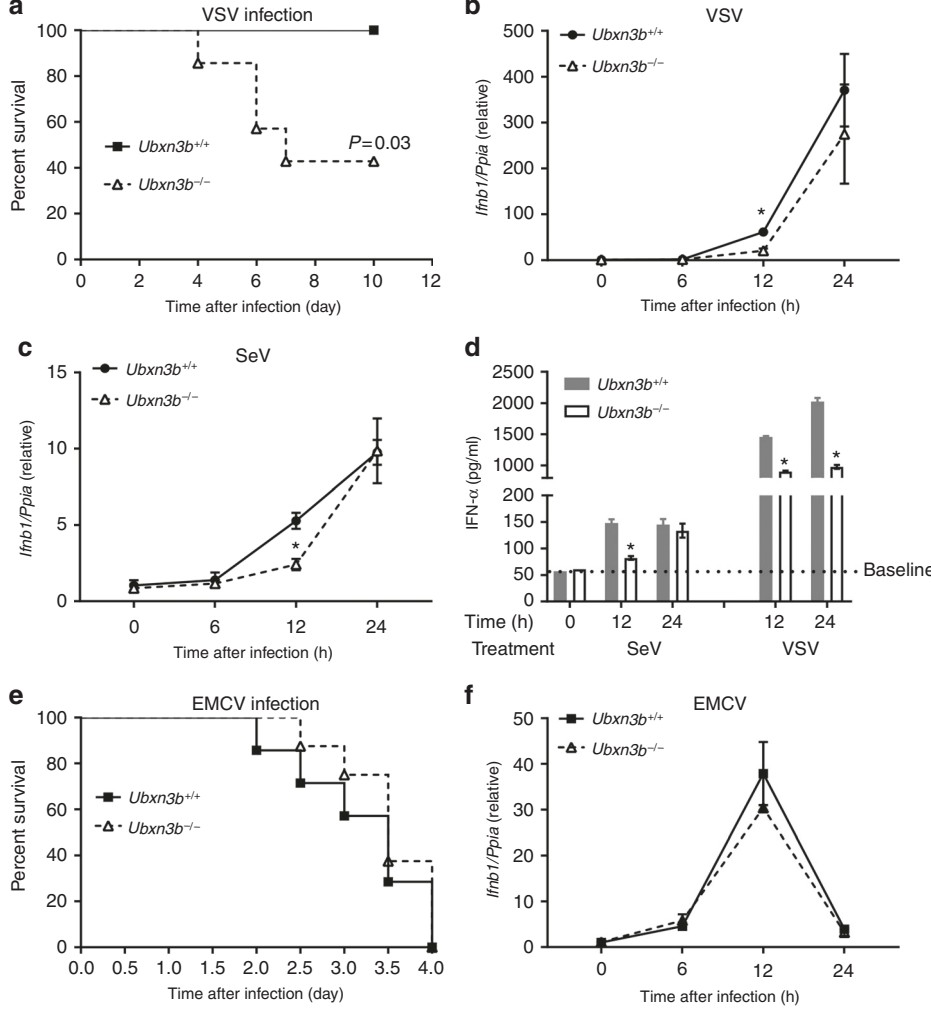

**Fig. 3** UBXN3B is critical for type I IFN responses to SeV and VSV, but not EMCV infection. **a** The survival curves of $Ubx3b^{-/-}$ ($N = 6$) and $Ubxn3b^{-/-}$ ($N = 7$) mice infected with $1 \times 10^7$ PFU of VSV i.v. The results are pooled from two experiments. *$P < 0.05$ (log-rank test). **b**, **c** qPCR analysis of $Ifnb1$ expression in cDCs infected with **b** VSV (MOI = 5) or **c** SeV (200 hemagglutination units per $10^5$ cells). **d** ELISA of IFN-$\alpha$ in the cell culture supernatants of cDCs infected with SeV and VSV as in **b**, **c**. $N = 3$ per genotype per time point. *$P < 0.05$ (unpaired Student's $t$ test). **e** The survival curves of $Ubx3b^{+/+}$ ($N = 7$) and $Ubxn3b^{-/-}$ ($N = 8$) mice infected with 200 PFU of EMCV i.p. $P = 0.53$ (log-rank test). The results are pooled from two experiments. **f** qPCR analysis of $Ifnb1$ expression in cDCs infected with EMCV (MOI = 5) for the indicated time. Bars/data points: mean ± s.e.m. Two biological replicates were pooled for qPCR ($N = 2$ per genotype per time point). *$P < 0.05$ (unpaired Student's $t$ test). The results are representative of two independent experiments

physiologically related to vertical transmission of HSV-1; (2) can be passaged in vitro for up to 12–15 generations[27], which allows us to knockout the genes of interest with CRISPR-Cas9 and examine their functions; and (3) can be obtained in a large quantity. In parallel, we included $STING^{-/-}$ as a positive control. As shown in Fig. 2h, STING and UBXN3B protein expression was depleted in knockout cells. Both $STING^{-/-}$ and $UBXN3B^{-/-}$ cells were more permissive to HSV-1 infection than WT (Fig. 2i); the

**Fig. 2** UBXN3B is critical for STING-dependent IFN-I induction in mouse primary cells. **a** ELISA of IFN-$\alpha$ in the cell culture supernatants of ($Ubxn3b^{+/+}$, $Ubxn3b^{-/-}$) bone marrow-derived macrophages (M$^\phi$), Flt3-induced pDCs, and GM-CSF-induced cDCs (pooled from five littermates) 20 h after the indicated treatments. $N = 3$ per genotype. *$P < 0.05$ (unpaired Student's $t$ test). **b** Immunoblots of an interferon-stimulated gene Oas1a, Sting, and Ubxn3b expression in cDCs 20 h after the indicated treatments. Tubulin is a housekeeping protein control. qPCR analysis of (**c**) $Ifnb1$ and $Tnfa$ mRNA expression and **d** cellular HSV-1 genome loads in cDCs infected with HSV-1 (MOI = 10) for the indicated time. **e** Immunoblots showing Ubxn3b and housekeeping Gapdh protein expression in mock ($Ubxn3b^{+/+}$) or 4-hydroxyl tamoxifen-treated Cre$^{+/-}$ $Ubxn3b^{flox/flox}$ ($Ubxn3b^{-/-}$) MEFs. **f** Viral titers (plaque-forming units/ml) in the supernatant of MEFs infected with HSV-1 (MOI = 0.1). $N = 3$ per genotype per time point. *$P < 0.05$; **$P < 0.01$ (unpaired Student's $t$ test). **g** qPCR analysis of selected immune gene mRNA expression in MEFs infected with HSV-1 as in **g**. **h** The immunoblots show knockout efficacy of STING and UBXN3B by CRISPR-Cas9 in human primary trophoblasts. Actin is a housekeeping protein control. **i** Fluorescent microscopic images of human primary trophoblasts infected with HSV-1-GFP (MOI = 0.3) for 18 h. Objective, ×5. Scale bar, 10 μm. **j** qPCR analysis of $Ifnb1$ mRNA expression in human primary trophoblasts infected with HSV-1-GFP for the indicated time. Bars/data points: mean ± s.e.m. Two biological replicates were pooled for qPCR ($N = 2$ per genotype per time point). *$P < 0.05$; **$P < 0.01$ (unpaired Student's $t$ test). The results are representative of two independent experiments

*IFNB1* mRNA expression was, however, much lower in knockout cells (Fig. 2j). These results demonstrate that the STING-regulating function of UBXN3B is evolutionarily conserved.

**UBXN3B regulates immune responses to some RNA viruses.** STING signaling is not only essential for induction of immune responses to DNA viruses but also important for antiviral immunity to certain RNA viruses such as VSV and Sendai virus (SeV). We then investigated the physiological role of UBXN3B during RNA virus infection. Consistent with the phenotype of *Sting*[−/−] infected with VSV[5], *Ubxn3b*[−/−] mice ($N = 6$) were also more susceptible to lethal VSV infection than its *Ubxn3b*[+/+] littermates ($N = 7$ each genotype, $P < 0.05$, log-rank test) (Fig. 3a). The *Ifnb1* mRNA expression was modestly decreased in knockout cDCs at 12 h after infection (Fig. 3b). Similarly the *Ifnb1* transcripts were also reduced in knockout cells infected with SeV (Fig. 3c). Consistent with this, the IFN-α protein concentration in the knockout cell supernatants was modestly lower than WT (Fig. 3d). However, Ubxn3b was dispensable for the control of a non-enveloped (+) single-stranded RNA virus, encephalomyocarditis virus (EMCV) infection in vivo (Fig. 3e), and for innate immune responses in cDCs (Fig. 3f). This is also consistent with the phenotype of *Sting*[−/−] mice[5].

**UBXN3B regulates STING signaling.** All the aforementioned in vivo and in vitro results clearly indicate an essential role of UBXN3B in the STING signaling pathway. To see whether UBXN3B also plays a role in other pathogen pattern recognition receptors such as RLR and TLR signaling pathways, we examined immune response induction in cDCs by several well-established

RLR/TLR agonists. The results show that *Ifnb1* and *Tnfa* mRNA upregulation by CpG DNA (TLR9), FLS-1 (Pam2CGDPKHPKSF, TLR2/6), lipopolysaccharide (LPS, TLR4), high-molecular-weight polyIC (TLR3, MDA5), and single-stranded poly-uridine (polyU, TLR7) in *Ubxn3b*[−/−] was not significantly different from *Ubxn3b*[+/+] cDCs (Supplementary Fig. 5a–e). These data demonstrate that UBXN3B is not essential for TLR/RLR-dependent signaling[28]. We next asked if UBXN3B regulated the IFN-JAK-STAT pathway, which induces expression of a large number of antiviral effectors. As shown in Supplementary Fig. 5f, the *Isg15* and *Oas1a* mRNA expression levels were similar between *Ubxn3b*[+/+] and *Ubxn3b*[−/−] cDCs, suggesting that UBXN3B is dispensable for the JAK-STAT signaling pathway. Furthermore, we observed that *Ubxn3b* expression was induced by IFN-α in both cDCs and MEFs, which was completely dependent on the IFN-I receptor signaling (Supplementary Fig. 6). Intriguingly, we noted that *Ubxn3b* was gradually and constantly upregulated throughout IFN-α treatment (Supplementary Fig. 6a). These data suggest that UBXN3B is an ISG.

We next explored the molecular mechanism by which UBXN3B acts on STING-dependent signaling. Lysine (K) 63-linked ubiquitination and dimerization of STING is a prerequisite for STING trafficking out of the ER to perinuclear vesicles where it recruits the antiviral kinase TBK1 to induce IFN-I[7,9,10,29]. STING is then phosphorylated and degraded via autophagy[29] and proteasomes[30]. UBXN proteins are likely involved in regulation of E3 ubiquitin ligases[20]. We first asked whether UBXN3B played a role in STING dimerization, trafficking, and phosphorylation. Indeed, STING dimerization took places 8 h after HSV-1 infection in *Ubxn3b*[+/+] MEFs, but this was obviously reduced in *Ubxn3b*[−/−] cells (Fig. 4a). We observed a similar phenotype in

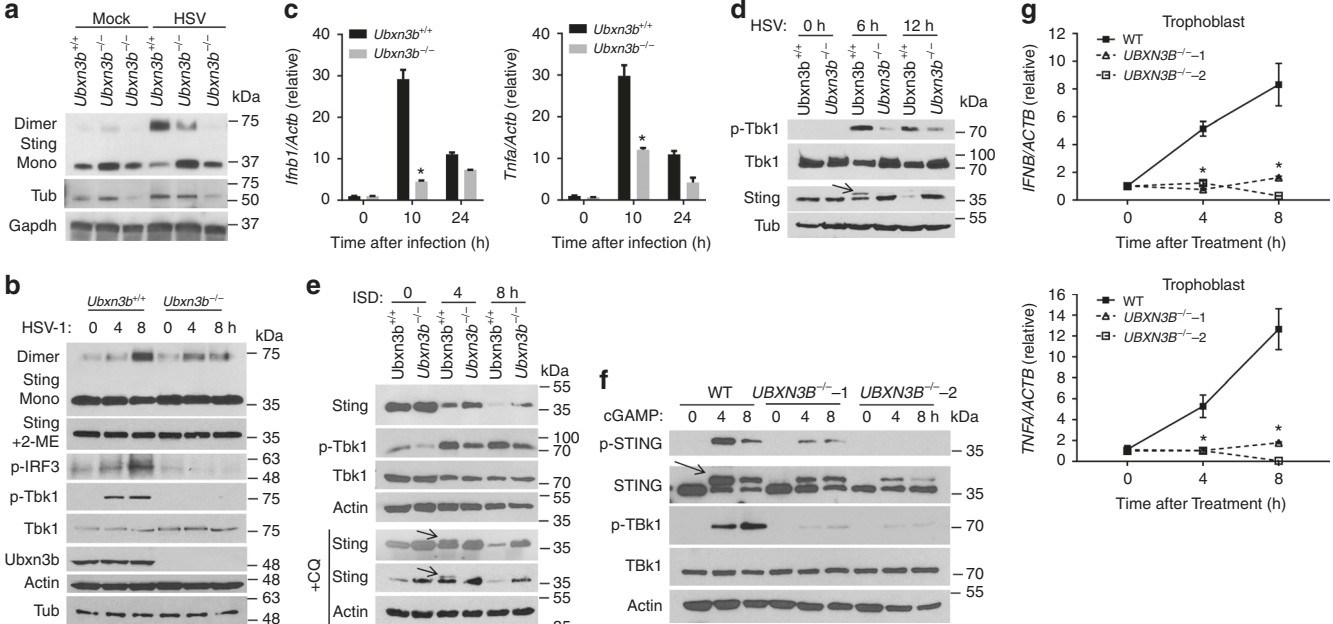

**Fig. 4** UBXN3B regulates STING dimerization, phosphorylation, and degradation. **a** Immunoblots showing Sting dimerization in untreated (*Ubxn3b*[+/+]) and 4-hydroxyl tamoxifen-induced Cre[+/−] *Ubxn3b*[flox/flox] (*Ubxn3b*[−/−]) primary MEFs. The cells were infected without (mock) or with HSV-1 (MOI = 0.5) for 8 h. **b** Immunoblotting analysis of the whole-cell lysates of bone marrow-derived cDCs infected with HSV-1 (MOI = 5). Mono monomer, 2-ME β-mercaptoethanol, a chemical compound that reduces disulfide bonds. **c** qPCR quantification of *Ifnb1* and *Tnfa* mRNA induction in cDCs by HSV-1 (MOI = 5). Bars: mean ± s.e.m. Two biological replicates were pooled for qPCR ($N = 2$ per genotype per time point). *$P < 0.05$ (unpaired Student's *t* test). **d–f** Immunoblotting analysis of the whole-cell lysates of **d** MEFs infected with HSV-1 (MOI = 0.5), **e** MEFs transfected with ISD (8 μg/ml) in the absence or presence of 40 μM of chloroquine (+CQ) and **f** trophoblasts transfected with cGAMP (8 μg/ml). In **e**, **f**, the arrows indicate phosphorylated STING with long and short exposure. Actin, Tubulin (Tub), and GAPDH are housekeeping protein controls. **g** qPCR quantification of *IFNB1* and *TNFA* mRNA induction in trophoblasts transfected with cGAMP (8 μg/ml). In **c**, **g**, the bars are: mean ± s.e.m. Two biological replicates were pooled for qPCR ($N = 2$ per genotype per time point). *$P < 0.05$ (unpaired Student's *t* test). The results are representative of two to three independent experiments

cDCs (Fig. 4b). Moreover, HSV-1-induced phosphorylation of TBK1 and IRF3 was inhibited in *Ubxn3b*[−/−] cDCs (Fig. 4b). Consequently, *Ifnb1* and *Tnfα* mRNA transcripts were also decreased in *Ubxn3b*[−/−] (Fig. 4c). STING is degraded after HSV-1 infection in permissive cells like MEFs. We extended the HSV-1 infection and indeed observed overt Sting degradation in *Ubxn3b*[+/+] cells at 12 h and this was inhibited in knockout cells (Fig. 4d). A similar result was noted in H1975 cells (Supplementary Fig. 7). ISD also stimulated STING degradation, which was partially blocked in *Ubxn3b*[−/−] (Fig. 4e). These data suggest that STING translocation from the ER to perinuclear vesicles is influenced by UBXN3B deficiency too. We investigated this in H1975 cells by immunofluorescence microscopy. In unstimulated cells, STING co-localized with an ER-resident protein calreticulin very well. STING was localized to some punctate structures that were away from calreticulin-stained ER 3 h after ISD stimulation in WT cells, but remained co-localized with calreticulin in ISD-treated *UBXN3B*[−/−] cells (Supplementary Fig. 8). Following translocation, STING is phosphorylated and degraded partially by autophagy. We did not readily detect phospho-Sting in ISD-treated MEFs, likely because of its rapid degradation following phosphorylation in these cells. We thus added chloroquine to block autophagy-mediated Sting degradation in order to capture

phospho-Sting. Indeed, chloroquine treatment delayed Sting degradation and Sting phosphorylation was observed at 4 h after ISD treatment in *Ubxn3b*[+/+] cells but clearly absent in *Ubxn3b*[−/−] MEFs (Fig. 4e). Interestingly, STING phosphorylation was much readily detected in human trophoblasts. Using a specific antibody against only the phosphorylated human STING, we noted rapid phosphorylation of STING at 4 h after cGAMP treatment. STING phosphorylation was obviously reduced, so was TBK1; while STING degradation was inhibited in *UBXN3B*[−/−] cells (Fig. 4f). As a result, *IFNB1* and *TNFA* mRNA upregulation by cGAMP was repressed in *UBXN3B*[−/−] cells (Fig. 4g).

The aforementioned data clearly demonstrate that UBXN3B regulates STING signaling and its downstream immune responses. To test if UBXN3B directly participated in STING signaling complex, we employed a co-immunoprecipitation assay in HEK293T cells with simultaneous expression of FLAG-tagged UBXNs and HA-tagged STING. UBXN3B but not its sibling UBXN3A co-immunoprecipitated with STING (Fig. 5a), suggesting that UBXN3B–STING binding is specific and that UBXN3B is involved in STING signaling. By fluorescence microscopy, we observed that UBXN3B co-localized with STING to the ER in unstimulated cells (Mock), and partly translocated together with STING to punctate structures after ISD treatment (Fig. 5b). We

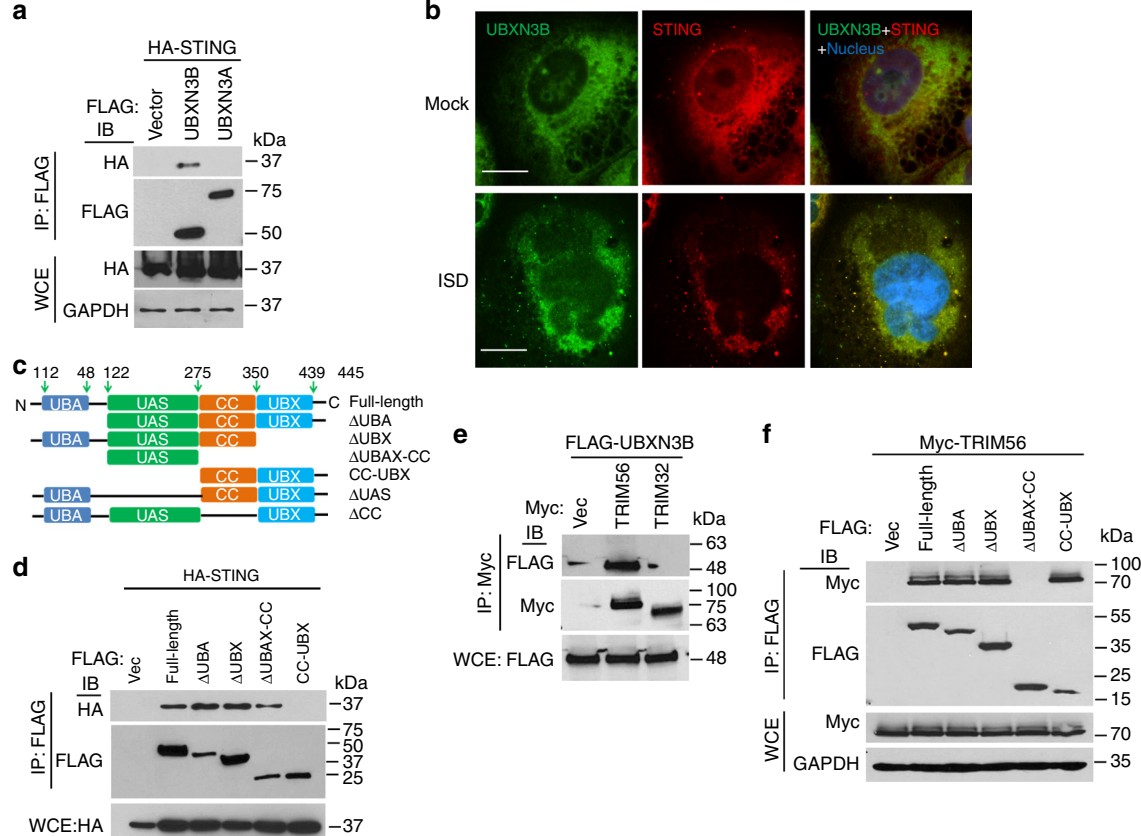

**Fig. 5** UBXN3B interacts with STING and TRIM56. **a** Co-immunoprecipitation (co-IP) of STING with UBXN3B from HEK293 cells transfected with FLAG-UBXNs and HA-STING plasmids using anti-FLAG magnetic beads, followed by immunoblotting (IB) with an anti-HA (STING) and FLAG (UBXN) antibody. **b** Immunofluorescence staining of STING and UBXN3B in H1975 cells treated without (Mock) or with ISD (8 μg/ml) for 3 h. The cells were stained with a mouse anti-human STING and rabbit anti-UBXN3B antibody followed by a secondary antibody conjugated with Alexa Fluor 594/488. The nuclei were stained with DAPI. Objective, ×40. Scale bar, 200 μm. **c** A schematic diagram of UBXN3B functional domains. **d** Co-IP of STING with the truncated forms of UBXN3B. The procedures are similar to **b**. **e** Co-IP of UBXN3B with TRIM from HEK293 cells transfected with FLAG-UBXN3B and Myc-TRIM plasmids using anti-Myc magnetic beads, followed by IB with an anti-Myc (TRIM) and FLAG (UBXN3B) antibody. **f** Co-IP of TRIM56 with the truncated forms of UBXN3B from HEK293 cells transfected with FLAG-UBXN3B truncates and Myc-TRIM56 plasmids using anti-FLAG magnetic beads, followed by IB with an anti-Myc (TRIM56) and FLAG (UBXN truncates) antibody. GAPDH is a housekeeping protein control. WCE whole-cell extract. The results are representative of two independent experiments

further showed that the UAS domain of UBXN3B was required for UBXN3B–STING interaction (Fig. 5c, d; Supplementary Fig. 9 a, b).

Ubiquitination of STING is essential for its downstream signaling and induction of antiviral immune responses. Several UBXNs including UBXN3B with the UBA-UBX domain are putative adaptor molecules that interact with E3 ligases and their substrates[20]. They likely also determine substrate specificity under different physiological conditions. We hypothesized that UBXN3B might serve as an adaptor that bridges STING and its E3 ligase. Three E3 ligases TRIM32, TRIM56, and AMFR have recently been shown to mediate STING ubiquitination during viral infection[7,8,15]. In particular, TRIM56 mediates both covalent and non-covalent ubiquitination of STING[7]. We first evaluated the ability of each of the E3 ligases to activate STING-dependent IFN-I responses. Results show that overexpression of TRIM56, but not AMFR, dramatically activated and synergized with cGAMP to induce STING-dependent IFN-I, while TRIM32 modestly activated and synergized with cGAMP to induce IFN-I (Supplementary Fig. 10). We next determined whether TRIM56 or TRIM32 associated with the STING–UBXN3B complex. UBXN3B co-immunoprecipitated with TRIM56 but not TRIM32 (Fig. 5e). The coil-coiled domain of UBXN3B mediated its

interaction with TRIM56 (Fig. 5f; Supplementary Fig. 9c). Very recently, another E3 ligase MUL1 was shown to be essential for STING ubiquitination on K224 and IRF3 activation[31]. We performed co-immunoprecipitation with MUL1 and UBXN3B overexpression in HEK293 cells to see if UBXN3B also regulated MUL1–STING interaction. We however observed no interaction between MUL1 and UBXN3B. These data suggest that UBXN3B specifically regulates STING–TRIM56 interaction. Indeed, HSV-1 infection induced endogenous Sting binding to Trim56 and Ubxn3b in MEFs (Fig. 6a). Furthermore, viral infection-induced Sting–Trim56 binding was almost abrogated in $Ubxn3b^{-/-}$ cells but was restored in UBXN3B-reconstituted knockout cells (Fig. 6b). These data overall demonstrate that UBXN3B is critical for HSV-1-induced STING–TRIM56 interaction. We next examined the role of UBXN3B in TRIM56-mediated STING ubiquitination. Overexpression of TRIM56 enhanced endogenous K63-linked ubiquitination of STING in WT cells (HEK293T-STING cell line), but this effect was reduced in $UBXN3B^{-/-}$ cells (Fig. 6d) ($UBXN3B^{-/-}$ was generated on HEK293T-STING background (see Fig. 6c). As previously described[7], K150R mutation decreased STING K63 ubiquitination by TRIM56 (Supplementary Fig. 11a). *Trans*-complementation of UBXN3B in knockout cells restored normal STING ubiquitination (Fig. 6d).

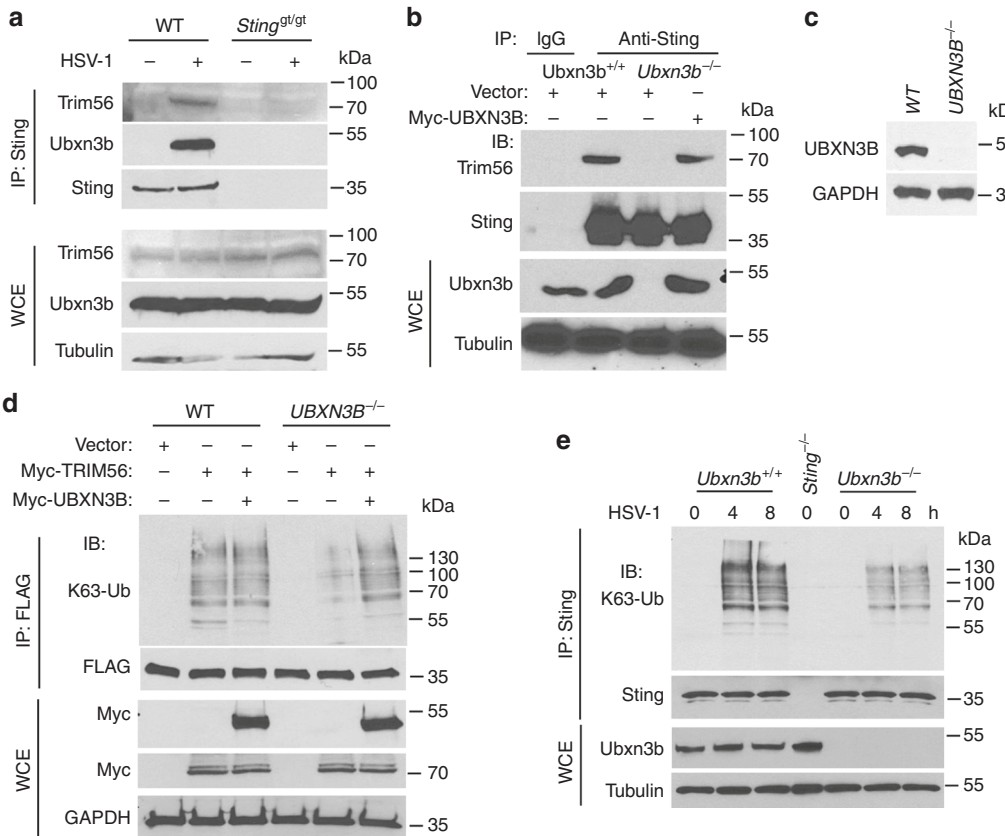

**Fig. 6** UBXN3B mediates STING interaction with and ubiquitination by TRIM56. **a** WT and *Sting*[gt/gt] MEFs were infected with or without HSV-1 (MOI = 0.5) for 8 h. Sting was immunoprecipitated (IP) from the MEF lysates using a rabbit anti-Sting antibody. The immunoblots indicate Sting co-immunoprecipitation with Ubxn3b and Trim56 after viral infection. WCE whole-cell extract. **b** WT and $Ubxn3b^{-/-}$ MEFs were transfected with vector or FLAG-UBXN3B plasmids by electroporation, and then infected with HSV-1 (MOI = 0.5) for 8 h. Sting IP was performed as in **a**. Rabbit IgG was used as a negative control. **c** The immunoblots show knockout efficacy of UBXN3B in HEK293T-STING cell line. **d** WT and $UBXN3B^{-/-}$ cells (parent cell line is HEK293T-FLAG-STING) were transfected with the indicated combinations of plasmids. FLAG-STING was immunoprecipitated (IP) with anti-FLAG magnetic beads. The proteins in IP and WCE were immunoblotted by an anti-K63-linked polyubiquitin, anti-Myc (UBXN3B and TRIM56) and anti-FLAG (STING) antibody, respectively. **e** Sting was precipitated with a rabbit anti-Sting antibody from cDCs infected without or with HSV-1 (MOI = 10) for 4 and 8 h. The $Sting^{-/-}$ cDCs were used as an IP control. The proteins in IP and WCE were immunoblotted with an anti-K63 and Sting antibody, respectively. Tubulin and GAPDH are housekeeping controls. The results are representative of two independent experiments

To validate the aforementioned overexpression results, we examined endogenous K63 ubiquitination of Sting in cDCs. HSV-1 infection induced K63 ubiquitination of Sting at 4 and 8 h after infection in $Ubxn3b^{+/+}$, but this was obviously impaired in $Ubxn3b^{-/-}$ cDCs (Fig. 6e). We recapitulated this in primary human trophoblasts treated with cGAMP (Supplementary Fig. 11b). Of note, co-immunoprecipitation of STING with TBK1 was observed in WT cells, but not in $UBXN3B^{-/-}$ cells, after cGAMP stimulation (Supplementary Fig. 11b). These data suggest that UBXN3B modulates TRIM56-mediated K63-linked ubiquitination of STING and activation of TBK1.

## Discussion

Being a central molecule of the anti-DNA virus immune signaling pathway, STING is subjected to complex regulation by cellular factors and also many viruses[12]. We here present physiological and biochemical evidence that UBXN3B controls DNA virus infection by regulating STING-mediated antiviral immune responses. First, ectopic expression of UBXN3B potentiates specifically STING-dependent IFN-I induction. Second, $Ubxn3b^{-/-}$ mice and cells are deficient in IFN-I induction by HSV-1, ISD, and cGAMP that stimulate STING signaling. Third, $Ubxn3b^{-/-}$ mice are highly susceptible to HSV-1 and VSV infection, but not to EMCV infection, recapitulating the $Sting^{-/-}$ phenotypes. Fourth, UBXN3B interacts with both STING and TRIM56 simultaneously, and HSV-1-induced STING–TRIM56 interaction is compromised in $UBXN3B^{-/-}$ cells. Fifth, UBXN3B deficiency leads to decreased K63 ubiquitination and dimerization of STING, and consequent phosphorylation of TBK1 and IRF3. STING translocation out of the ER to the ER–Golgi intermediate compartment, followed by phosphorylation and degradation after HSV-1/ISD/cGAMP stimulation, is also partly blocked in $UBXN3B^{-/-}$ cells. Lastly, UBXN3B can be induced by IFN-I, likely to form a positive feedback loop to stimulate antiviral immune responses. All these observations strongly suggest that UBXN3B directly regulate STING signaling. However, STING-mediated immune responses are not completely abolished in $UBXN3B^{-/-}$ cells and mice, suggesting that UBXN3B-independent regulation of STING and IFN-I-inducing pathways exist. Several E3 ligases that may not be directly relevant to UBXN3B have been shown to positively regulate STING signaling[8,15,31]. Moreover, in pDCs TLR9 contributes to induction of the IFN-I responses by HSV-1[32,33], which however is not regulated by UBXN3B.

The UBA-UBX-containing UBXN proteins may be adaptors that bridge multiple E3 ligases and a variety of substrates[20], and likely essential for embryonic development. Indeed, by using different genome editing technologies, we and collaborators independently find that $Ubxn3b$-null mutation is embryonically lethal[24]. In agreement with our findings, another UBA-UBX gene, $UBXN1$, is also essential for embryonic development[34]. The genes essential for embryonic/neonatal stages are generally believed to be essential for adult stage too. For example Gpx4 and p53 is not only essential for embryos/neonates but also for adult mice[25,26]. Surprisingly, adult mice depleted of $Ubxn3b$ survive and develop as normally as WT mice. This interesting finding suggests that some of the genes indispensable for embryonic development may become non-essential at later stages of development. Our results provide proof of principle for in-depth study of the physiological functions of essential genes like $UBXN3B$ in adult systemic knockout mice.

Of note, Ubxn3b protein expression is still abundant and ubiquitous in adult mice, suggesting that it plays different physiological roles than that at the embryonic stage. Previous studies demonstrated that UBXN3B regulated triglyceride synthesis and lipid droplet turnover in vitro cell cultures[22,23], and promoted very low-density lipoprotein secretion out of hepatocytes in mice fed with a high-fat diet[24]. More recently, UBXN3B was shown to regulate de novo sterol biosynthesis by targeting the rate-limiting enzyme of the mevalonate pathway-HMG-CoA reductase to proteasomal degradation[35]. In the current study, we show that under viral infection conditions, UBXN3B bridges the ER protein STING and its cytoplasmic E3 ligase TRIM56 to initiate antiviral immune responses. The multifaceted nature of UBXN3B is likely attributable to its capability as an adaptor to interface different E3 ligases[20] and substrates under different physiological conditions. The involvement of an adaptor in an E3 ligase action could greatly expand the substrate spectrum of the E3 ligase in response to various physiological and environmental cues.

Following DNA virus infection, STING undergoes four forms of polyubiquitination, K11-linked[36], K27-linked[15], K48-linked[16,17], and K63-linked[7,8,31], resulting in contrasting effects on STING-mediated immune signaling. Particularly, K27-linked and K63-linked polyubiquitination activates STING. In agreement with published data[7,8], forced expression of K63-E3 ligases TRIM32 and TRIM56 enhanced STING-dependent IFN-I induction and significantly potentiated the cGAMP effects on STING. However, overexpression of a K27-E3 ligase AMFR failed to induce IFN-Is by itself or together with cGAMP, suggesting a different mechanism of AMFR action than TRIM32 and TRIM56. A similar phenomenon has been observed for TRAF3, a K63-E3 ligase critical for TLR-dependent and RLR-dependent IFN-I induction[37]. It is not surprising that redundant K63-E3 ligases are available for STING, since it is crucial for induction of anti-DNA virus immunity. Indeed, recently another K63-E3 ligase MUL1 was shown to ubiquitinate the K224 of STING and specifically regulate IRF3 activation, followed by STING phosphorylation and degradation[31]. Our results demonstrate that UBXN3B regulates TRIM56-mediated ubiquitination of STING. It binds STING via the UAS domain and TRIM56 via the coiled-coil domain, likely serving as an adaptor to bring STING to the vicinity of TRIM56. K63-linked polyubiquitination of STING can be in a covalent form, which is necessary for STING oligomerization and activation of downstream immune responses[7]. Like RIG-I, STING could also be non-covalently modified with and activated by unanchored polyubiquitin chains[7,37]. In this scenario, UBXN3B could play a role by binding and bringing polyubiquitin chains to STING, in light of recent findings that the UBA domain of some UBXNs was able to bind polyubiquitin chains[20]. It is thus possible that UBXN3B could also in part regulate STING ubiquitination catalyzed by other E3 ligases indirectly.

In summary, we present physiological and biochemical evidence that UBXN3B controls DNA virus infection by regulating STING signaling. Our data offer an insight into the previously unappreciated immune regulatory function of UBXN3B and provide proof of methodology for investigating the physiological functions of certain essential genes like $UBXN3B$ in adult systemic knockout animals.

## Methods

**Mouse models**. Mice with the exon 1 of Ubxn3b flanked by two LoxP sites ($Ubxn3b^{flox/flox}$) were generated via homologous recombination by Dr. Fujimoto at Nagoya University[24]. The homozygous $Ubxn3b^{flox/flox}$ were then crossed with homozygous tamoxifen-inducible Cre recombinase-estrogen receptor T2 mice (The Jackson Laboratory, Stock # 008463) to generate male and female $Cre^{+/-}$ $Ubxn3b^{flox/flox}$ littermates, which were mated to produce $Cre^{+/-}Ubxn3b^{flox/flox}$, $Ubxn3b^{flox/flox}$, $Cre^{+/-}$ for experimentation. Genotyping PCR was performed with ~40 ng genomic DNA and Choice Taq Blue Mastermix (Denville Scientific, Cat# CB4065-8) under the following cycling conditions: 94 °C for 2 min (94 °C for 1 min, 60 °C for 30 s, 72 °C for 30 s) ×34 cycles, 72 °C for 7 min[24]. The primers for LoxP were: forward primer, 5′-CCAACCTCTGTGGGTCCTC-3′ and reverse primer, 5′-GCGATTGGGGGATCTGAGTTA-3′. This PCR reaction resulted in a product of 187 bp (WT, LoxP negative) and/or 245 bp (mutant, LoxP positive). The

primers for genotyping Cre were: common-5′AAGGGAGCTGCAGTGGAGTA-3′, WT reverse, 5′-CCG AAA ATC TGTGGGAAGTC-3′, and mutant reverse, 5′-CG GTTATTCAACTTGCACCA-3′. The PCR reaction resulted in a product of ~300 bp (WT, Cre negative) and/or 450 bp (mutant, Cre positive). To induce Ubxn3b deletion, weaned mice were treated with 100 µl of tamoxifen (10 mg/ ml in corn oil) (Sigma, #T5648) by intraperitoneal injection every 2 days for a total duration of 8 days[26]. The treated mice were housed for 1–2 weeks, allowing tamoxifen to be metabolized completely (half-life ~16 h). The Sting-deficient strain (Sting[gt/gt]) was initially developed by Dr. Russell Vance at the University of California, Berkeley[38], and is now available at the Jackson Laboratory (Stock # 017537). Sting[−/−] (null knockout of exons 3–5) (Stock # 025805) and IFN-I receptor knockout (Ifnar1[−/−]) (Stock # 028288) mice were also from the Jackson Laboratory. We used 8–10-week-old sex-matched (both female and male) mice for all the experiments. All the animal protocols were approved by the Institutional Animal Care and Use Committee at New York Medical College and Yale University adhering to the National institutes of Health recommendations for the care and use of laboratory animals.

**Reagents and antibodies**. The rabbit anti-GAPDH (D16H11, Cat# 5174, 1:3000), calreticulin (D3E6, Cat# 12238, 1:200 for IFA), α-Tubulin (Cat# 2144, 1:3000), UBXN3B (D8H6D, Cat# 34945, 1:1000 for IB, 1:100 for IFA), TBK1 (D1B4, Cat# 3504, 1:1000), phospho-TBK1 (D52C2, Cat# 5483, 1:500), K63-linked poly-ubiquitin (D7A11, Cat# 5621,1:500), phospho-IRF3 (4D4G, Cat# 4947, 1:500), human phospho-STING (Cat# 85735, 1:500), and STING (D2P2F, Cat#13647, 1:500) were purchased from Cell Signaling Technology (Danvers, MA, USA). The goat anti-TRIM56 (E-18, Cat# sc-249085, 1:200) was obtained from Santa Cruz Biotechnology (CA, USA). The goat anti-UBXN3B was obtained from Genway Biotech (Cat# GWB-5E6F34, 1:500, San Diego, CA, USA). The mouse anti-FLAG (OTI4C5, Cat# TA50011, 1:2000), Myc (9E10, Cat# TA150121, 1:1000), and HA (HA.C5, Cat# TA100012, 1:1000) were obtained from Origene (Rockville, MD, USA). Lipofectamine 2000 was obtained from ThermoFisher (Waltham, MA, USA). 2′3′-cGAMP (Cat# tlrl-nacga23), IFN stimulatory DNA derived from *Listeria monocytogenes* genome (ISD, Cat# tlrl-isdn), polyI:C (1.5–8 kb, Cat# tlrl-pic), FSL-1 (Pam2CGDPKHPKSF, Cat# tlrl-fsl), CpG DNA (5′-tcgtcgtttttcggcgc:gcgccg-3′, Cat# tlrl-2395), LPS (Cat# tlrl-pb5lps), and ssPolyU (Cat# tlrl-sspu) were products of Invivogen (San Diego, CA, USA). The anti-FLAG M2 magnetic beads (Clone M2, Cat# M8823) were from Sigma-Aldrich (St. Louis, MO, USA) and anti-Myc magnetic beads (Clone 9E10, Cat# 88842) were from ThermoFisher (Waltham, MA, USA). The mouse anti-human STING (Clone 723505, Cat# MAB7169, 1:200 for IFA) was available at R&D Systems (Minneapolis, MN, USA). Recombinant mouse IFN-α (Cat# 752802) was a product of BioLegend (San Diego, CA, USA). The rabbit anti-HSV-1/HSV-2 serum, which is specific for the ICP's and late structural (virion) antigens, was purchased from Antibodies-online (Cat# ABIN 285517, 1:500).

**Plasmid construction**. FLAG-Myc-UBXN2A (Cat# MR203340, NCBI accession: NM_145441) and FLAG-Myc-UBXN2B (Cat# MR204884, NCBI accession: NM_026534) on pCMV6-entry vector were purchased from Origene (Rockville, MD, USA). FLAG-UBXN1 was constructed by inserting the human UBXN1 open reading frame (ORF) (NCBI accession: NM_015853.4) into pcDNA3-FLAG/Myc vector[39]. The additional FLAG-UBXNs were a kind gift of Dr. Raymond J. Deshaies at California Institute of Technology, USA[20]. HA-STING and FLAG-STING-K150R were recently reported[7,40]. The truncated forms of human UBXN3B, human AMFR, TRIM32, and TRIM56 were amplified by PCR, cut by restriction enzymes, and inserted into pcDNA3-FLAG/Myc vector.

**Cell culture and viruses**. EMCV (Cat# VR129-B), SeV (Cantell strain, Cat# VR-907), VSV (Indiana strain, Cat# VR-1238), and HSV-1 (Cat# VR1493) were purchased from American Type Culture Collection (ATCC) (Manassas, VA, USA) and the multiplicity of infection (MOI) was specified in each figure legend[41]. Green fluorescent protein-tagged herpes simplex virus type I (KOS strain) was provided by Dr. Rongtun Lin at the McGill University[28]. All viruses were propagated in Vero cells.
HEK293T cells (Cat# CRL-3216), Vero cells (monkey kidney epithelial cells, Cat# CCL-81), H1975 cells (human lung epithelial cells (Cat# CR5908), and L929 cells (mouse fibroblast cells, Cat# CCL-1) were purchased from ATCC (Manassas, VA, USA). TLR4/MD2/CD14-expressing HEK293 cell line was obtained from Invivogen (Cat# 293-htlr4md2cd14, San Diego, CA, USA). HEK293T/Vero cells and L929/H1975 cells were grown in Dulbecco's modified Eagle's medium or Roswell Park Memorial Institute (RPMI) 1640, respectively (Life Technologies, Grand Island, NY, USA) supplemented with 10% fetal bovine serum (FBS) and antibiotics/antimycotics. These cell lines are not listed in the database of commonly misidentified cell lines maintained by ICLAC, and have not been authenticated in our hands. They are routinely treated with MycoZAP (Lonza) and tested for mycoplasma contamination in our hands.

**Generation of HEK293T-STING cell line**. Since HEK293T expresses a very low level of STING, we thus constructed a FLAG-STING-expressing stable cell line. The human STING ORF was subcloned into pcDNA3-FLAG vector, and then transfected into HEK293T cells using Lipofectamine 2000. FLAG-STING-expressing cells were selected with 200 µg/ml zeocin.

**Ubxn3b deletion in primary MEFs**. Pregnant Cre[+/−]Ubxn3b[flox/flox] females (mated to Cre[+/−] Ubxn3b[flox/flox] male) were euthanized on day 14 of gestation. Embryos were decapitated and eviscerated, and then digested with trypsin for 10 min at 37 °C rotating. Fibroblasts were filtered through a 100 µM filter, cultured in RPMI 1640 medium (Life Technologies, NY, USA), supplemented with 10% FBS and antibiotics/antimycotics, propagated for two passages, and then frozen[41,42]. Cre[+/−] Ubxn3b[flox/flox] MEFs were identified by genotyping. The Cre[+/−] Ubxn3b[flox/flox] MEFs were treated with 4-hydroxyl tamoxifen at 0.01 mg/ml for 3 days to generate Ubxn3b[−/−] cells. After induction, the cells were further passaged two times in RPMI medium. Non-treated Cre[+/−] Ubxn3b[flox/flox] MEFs were considered Ubxn3b[+/+].

**Differentiation of BMDMs and DCs**. BMDMs were differentiated using a published method[43]. Briefly, bone marrow cells were isolated from mock-pretreated and tamoxifen-pretreated Cre[+/−] Ubxn3b[flox/flox] mice and then differentiated into macrophages in L929-conditioned medium (RPMI 1640, 20% FBS, 30% L929 culture medium, 1× antibiotics/antimycotics) in 10-cm Petri dishes at 37 °C, 5% $CO_2$ for 5 days. The culture medium was replaced by fresh L929-conditioned medium every 2 days. The attached BMDMs were dislodged by pipetting and counted for plating in 12-well or 6-well cell culture plates. Bone marrow-derived cDCs and pDCs were induced from bone marrow cells with 10 ng/ml murine GM-CSF and 100 ng/ml Flt3L (PeproTech), respectively, for 6–8 days[44]. BMDMs/DCs were cultured in RPMI 1640 medium containing 10% (volume/volume (v/v)) FBS (Invitrogen, Carlsbad, CA, USA), 100 U/ml penicillin and 100 µg/ml streptomycin (Invitrogen) and maintained at 37 °C and 5% $CO_2$ overnight, and then washed once with pre-warmed fresh medium.

**Isolation of human trophoblasts**. Human trophoblasts were isolated from first trimester or term placenta using published methods[45,46]. Briefly, a piece of placenta was minced with a razor blade and then digested with 0.25% trypsin and 0.2 mg/ml DNAse I in Hank's balanced salt solution to release cells. The dispersed cells were passed through a 40 µm cell strainer, and trophoblasts were isolated through Percoll gradient (10–70% v/v) centrifugation. Cells were maintained in complete RPMI 1640 medium with 10% FBS and antibiotics. The IRB of Yale University and New York Medical College approved the study and deemed exemption of human subject study[47]. The study used de-identified leftover specimens which were otherwise discarded. The investigators did not have any contact with the subjects and had no access to the patient information and did not link the results to the subjects. The investigation was irrelevant to clinical care and uses random specimens. The research did not investigate the health/disease status/genetic information of the subjects.

**Ligand treatment and viral infection conditions**. PRR ligands were transfected into cells with Lipofectamine 2000 using standard procedures. For hard-to-transfect cells such as trophoblasts, H1975, MEFs, and DCs, transfection was carried out in cell suspension. Briefly, cells were dislodged by trypsin digestion and pelleted by brief centrifugation. The cell pellet was then suspended in the transfection mix (DNA + Lipofectamine 2000 prepared as above) for 20 min with intermittent gentle agitation. Pre-warmed RPMI 1640 complete medium was then added and plated for further culture. For viral infection, viruses were allowed to attach and infect cells for 2 h; the cells were then washed with pre-warmed 1× phosphate-buffered saline (PBS) once and incubated with fresh medium. The MOI and infection time were specified for each experiment in the figure legend.

**Generation of gene knockout cell lines with CRISPR-Cas9 technology**. Two target guide sequences (1-GGTCAGTGACCCGGCTGCGA, 2-TACGTTCCCTGGTAGAAGAC common for both mouse and human UBXN3B) were cloned into lentiCRISPRv2 vector and co-transfected into HEK293T cells with the packaging plasmids pVSVg and psPAX2[48,49]. The guide RNAs for human STING were: 1-GGTGGAGCACCAGGTAC, 2-GGTACCGGAGAGTGTGCTC). HEK293T-STING cells, trophoblasts, and H1975 cells were then transduced by lentiviral particles. The WT control was lentiCRISPRv2 vector only. The transduced cells were selected with puromycin at 1 µg/ml for trophoblasts, 1 µg/ml for HEK293T cells and 0.5 µg/ml for H1975 cells. Successful knockout clones were confirmed by immunoblotting.

**Plaque-forming assay**. Quantification of infectious viral particles in tissues/cell culture supernatant was performed on Vero cell monolayer[43,50]. Briefly, 30–100 µg (total proteins) of tissue lysates triturated in sterile PBS or serial dilutions of supernatant was applied to confluent Vero cells (6-well plate) at 37 °C for 2 h. The inoculum was then removed and replaced with 2 ml of DMEM complete medium with 1% SeaPlaque agarose (Cat# 50100, Lonza). Plaques were visualized using Neutral red (Sigma-Aldrich) after 3 days of incubation at 37 °C, 5% $CO_2$.

**Mouse infection and disease monitoring**. Six-week-old to eight-week-old sex-matched littermates were infected with $1 \times 10^7$ plaque-forming units (PFUs) of HSV-1/VSV intravenously or 200 PFU of EMCV intraperitoneally. Morbidity and mortality was monitored twice a day. Neurological symptoms were assessed using an arbitrary scoring criteria from 1 to 5 (where 1 indicated ruffled fur and hunched posture but can easily be made to move around; 2 indicated a hunched posture and slow to move; 3 indicated a hunched posture, some movement, and labored breathing; 4 indicated a hunched posture, labored breathing, and little or no movement; and 5 indicated moribund or dead)[51]

**Reverse transcription and qPCR**. A few grams of animal tissues, 25 µl of whole blood, and up to $1 \times 10^6$ culture cells were collected in 350 µl of RLT buffer (QIAGEN RNeasy Mini Kit). Soft tissues were homogenized using an electric pellet pestle (Kimble Chase LLC, USA). RNA was extracted following the QIAGEN RNeasy manufacturer's instructions. Reverse transcription of RNA into complementary DNA (cDNA) was performed using the BIO-RAD iScript™ cDNA Synthesis Kit. Quantitative PCR (qPCR) was performed with gene-specific primers and 6FAM–TAMRA (6-carboxyfluorescein–$N,N,N,N$-tetramethyl-6-carboxyrhodamine) probes or SYBR Green. Results were calculated using the $-\Delta\Delta$Ct method and a housekeeping gene as an internal control. The qPCR primers and probes for immune genes were reported in our previous studies[42,52]. The primer pair for mouse Ubxn3b was 5′-GAGAAATATGGGAGGGCACA-3′ and 5′-AGAA CAAGCCCAGAAAAGCA-3′. The primers for HSV-1 were: 5′-AGCCTGTACC CCAGCATCAT-3′ and 5′-ACCTCGATCTCCAGGTAGTCC-3′. The Taqman gene expression assays for Ifit1 (Mm00515153_m1), Oas1a (Mm00836412_m1), Isg15 (Mm01705338_s1), IFNB1 (Hs01077958_s1), and TNFA (Hs00174128_m1) were obtained from ThermoFisher Scientific. The housekeeping gene controls were beta actin (ACTB), peptidylprolyl isomerase A (PPIA), and hydroxymethylbilane synthase (HMBS). A previous study demonstrated that HMBS was the least unaffected housekeeping gene during HSV-1 infection; PPIA was most stable for other viral infections[53]. Human HMBS primers are: 5′-ACGGCTCAGATAGCA TACAAGAG-3′ and 5′-GTTACGAGCAGTGATGCCTACC-3′; mouse Hmbs: 5′-GTGCCTACCATACTACCTCCTG-3′ and 5′-ACTCTCCTCAGAGAGCTGGT TC-3′; human PPIA: 5′-GGCAAATGCTGGACCCAACACA-3′ and 5′-TGCTGG TCTTGCCATTCCTGGA-3′; mouse Ppia: 5′-CATACAGGTCCTGGCATCTTG TC-3′ and 5′-AGACCACATGCTTGCCATCCAG-3′. These primer sequences were available at Origene Technologies (Rockville, MD, USA).

**ELISA**. A commercial ELISA (enzyme-linked immunosorbent assay) Kit was used to measure the levels of IFN-α (Cat# 42120) and IFN-β (Cat# 42400, PBL Assay Sciences, Piscataway Township, NJ, USA) proteins in either cell culture supernatants or mouse sera.

**Immunoblotting**. Standard sodium dodecyl sulfate-polyacrylamide gel electrophoresis, Western blotting, and an enhanced chemiluminescent (ECL) substrate (ThermoFisher, Cat# 32106) was applied except for STING dimerization assay where no reducing agent was applied to the samples. In some cases, such as K63-Ub, an ultra-sensitive ECL substrate for low-femtogram-level detection (ThermoFisher, Cat# 34095) was used.

**Co-immunoprecipitation**. HEK293T cells were transfected with expression plasmids using Lipofectamine 2000. Whole-cell extracts were prepared from transfected cells in lysis buffer (150 mM NaCl, 50 mM Tris, pH 7.5, 1 mM EDTA, 0.5% NP40, 10% glycerol) and were incubated with 50 µl of anti-FLAG magnetic beads for 2 h at 4 °C. Co-immunoprecipitation was performed according to the manufacturer's instructions (anti-Flag magnetic beads, Sigma-Aldrich)

**Co-immunoprecipitation of endogenous proteins**. A total of $1 \times 10^7$ cells were lysed in 2 ml of lysis buffer (150 mM NaCl, 50 mM Tris, pH 7.5, 1 mM EDTA, 0.5% NP40, 10% glycerol, protease inhibitor cocktail). Lysates were cleared by centrifugation at $6000 \times g$ for 10 min at 4 °C. For UBXN3B trans-complementation, MEFs were transfected with 5 µg vector or FLAG-UBXN3B per $1 \times 10^6$ cells, respectively, by electroporation (Lonza, Cat# DCR1003) 20 h before HSV-1 infection. Two micrograms of rabbit IgG or 50 µl of anti-STING/per sample (Cell Signaling Technology, Cat# 13647) was cross-linked to 50 µl of protein A/G agarose beads (ThermoFisher, Cat# 20421) with dimethyl pimelimidate (ThermoFisher, Cat# 21666). The cleared lysates were incubated with the agarose beads with gentle agitation at 4 °C overnight. The beads were washed five times in ice-cold wash buffer (150 mM NaCl, 50 mM Tris, pH 7.5, 1 mM EDTA, 0.5% NP40), and bound proteins were eluted by boiling for 3 min in SDS sample lysis buffer.

**Dual luciferase reporter assays**. Seventy percent confluent HEK293T cells were transfected with 50 ng of pRL-TK reporter (herpes simplex virus thymidine kinase promoter-driven renilla luciferase; internal control), 100 ng of pGL3-ISRE luciferase reporter (firefly luciferase; experimental reporter) plasmid, and other expressing plasmids as appropriate. Twenty-four hours after transfection, luciferase activity was measured using a Promega Dual Glow Kit (Cat# E2980) according to the manufacturer's instructions.

**Immunofluorescent microscopy**. Cells were fixed with 4% paraformaldehyde. The cells were sequentially permeabilized with 0.5% Triton X-100, blocked with 2% goat serum, incubated with primary antibody (rabbit anti-UBXN3B, 1:100, Cell Signaling (Cat# 34945, Clone D8H6D); mouse anti-STING, 1:100, R&D Systems (Clone 723505, Cat# MAB7169); and rabbit anti-calreticulin, 1:200, Cell Signaling (Clone D3E6, Cat# 12238)) at 4 °C overnight, washed briefly, and then incubated with Alexa Fluor 488/594-conjugated goat anti-rabbit/mouse IgGs (1:200, Life Technologies, Cat# A11070) for 1 h at room temperature. Nuclei were counterstained with DAPI (4′,6-diamidino-2-phenylindole). Images were acquired using a Zeiss Axiovert-200 fluorescence microscope (objective, ×40).

**Graphing and statistics**. The sample size chosen for our animal experiments in this study was estimated based on our prior experience of performing similar sets of experiments and power analysis calculations (http://isogenic.info/html/ power_analysis.html). All animal results were included and no method of randomization was applied. For all the bar graphs, data were expressed as mean ± s.e.m. Prism 7 software (GraphPad Software) was used for survival curves, charts, and statistical analyses. Survival curves were analyzed using a log-rank (Mantel–Cox) test. For in vitro results, a standard two-tailed unpaired Student's t test was used. For animal studies, an unpaired two-tailed non-parametric/parametric Mann–Whitney U test was applied to statistical analysis. P values ≤0.05 were considered significant. The sample sizes (biological replicates), specific statistical tests used, and the main effects of our statistical analyses for each experiment were detailed in each figure legend.

**Data availability**. The datasets generated during and/or analyzed during the current study are available from the corresponding author upon request.

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

## Acknowledgements

This work was supported by a National Institutes of Health grant R01AI132526 to P.W., U19 AI089992 and R01 AI127865 to E.F. E.F. is also an investigator of the Howard Hughes Medical Institute.

## Author contributions

L.Y. performed the majority of the experimental procedures. G.Y., L.W., S.C., J.M., H.K., T.G., and D.G.M. contributed to some of the experiments. T.F., G.C., F.Y., R.L., and E.F. contributed to data analysis and/or provided technical support. L.Y. and P.W. conceived and designed; P.W. oversaw the study. L.Y. and P.W. wrote the paper and all the authors reviewed and/or modified the manuscript.

## Additional information

**Competing interests:** The authors declare no competing interests.

