## [Peer Review File · Nature Communications]

Reviewers' comments:

Reviewer #1 (Remarks to the Author):

In this work Yang et al have examined the role of ubiquitin regulatory X domain containing proteins in the STING signaling pathway. They find that UBXN3B is essential for STING signaling and for resistance against HSV-1 infection in mice. At the mechanistic level, the authors suggest that UBXN3B interacts with STING and TRIM56, thus promoting K63-linked ubiquitination of STING and downstream signaling. The work is well designed and the data are clear. However, the mechanistic data are still somewhat underdeveloped and it is also not clear whether the phenomenon presented applies in primary human cells:

MAJOR POINTS

1. Figure 2. The murine KO cells should be stimulated through a range of PRRs and cytokine receptors and gene expression patterns should be evaluated. It is essential to characterize the specificity of UBXN3B in immune signaling
2. Figure 3. For comparison IFN induction by a panel of RNA viruses should be tested in wt versus UBXN3B cells.
3. The mechanistic data are somewhat superficial. As a minimum, data should be presented on phospho-TBK1 and phospho-STING in wt versus UBXN3B cells
4. It is very surprising (and somewhat worrying) that none of the STING blots show the characteristic phospho-STING band. Can the authors explain this? Along the same lines, the authors also fail to observe STING degradation following stimulation. This is also in direct contrast to most – if not all – papers on STING signaling in the literature.
5. The work would gain significantly, and could potentially compensate for the lack of detailed mechanism, of data were provided on the role of UBXN3B in primary human cells.

Reviewer #2 (Remarks to the Author):

In this study, the authors generated inducible UBXN3B knockout mice and showed that the absence of UBXN3B leads to high lethality upon HSV-1 infection, which is correlated with decreased type I IFN (IFN-I) production. Moreover, HSV-1 and STING ligand-induced IFN-I was also repressed in *Ubxn3b*^{-/-}

primary cells, indicating that UBXN3B is involved in STING dependent pathways. The authors further explored the mechanism and showed that UBXN3B interacted with both STING and one of its E3 ligases, TRIM56, to facilitate STING ubiquitination and dimerization. Most of the experiments are well done. However, there are some experimental concerns that are listed below.

1. STING has been shown to be involved in RIG-I dependent RNA sensing pathway as well. Does RNA virus infection e.g. VSV or SeV infection have an effect on IFN responses in *Ubxn3b*^{-/-} mice compared to WT mice?
2. Figure 4. The authors should show HSV-1 or dsDNA-induced STING ubiquitination in WT and UBXN3B^{-/-} H1975 cells. Does UBXN3B collaborate with MUL1, another E3 ligase that ubiquitinates STING?
3. UBXN3B has been shown to interact with STING under ectopic expression conditions. It would be beneficial if the authors could also show the interaction of UBXN3B and STING under endogenous conditions. Is this interaction altered upon DNA stimulation conditions, such as HSV-1 infection, cGAMP or poly I:C transfection?
4. The authors have shown the interaction between STING and UBXN3B. Do UBXN3B and STING also co-localize in cells? If so, does the localization of UBXN3B and STING change upon DNA stimulation?
5. The authors propose that UBXN3B is an adaptor of TRIM56 and STING, therefore affecting K63-linked ubiquitination of STING by TRIM56. As TRIM56 has been shown to be critical for TBK1 recruitment to STING, is the TBK1-STING interaction affected in UBXN3B deficient cells and does this impair TBK1 phosphorylation and STING translocation?
6. A recent publication reported that K63-linked ubiquitination is required for dsDNA-induced STING translocation, phosphorylation and degradation (Sci Immunol. 2017). Considering that UBXN3B deficiency markedly reduced K63-linked ubiquitination of STING, the authors should also check whether UBXN3B deficiency affects STING trafficking, phosphorylation and degradation in response to dsDNA stimulation.
7. All figure legends need to indicate how many times the experiment was done and statistical significance needs to be shown for all experiments.

Minor:

1. Figure 1B and 1C. Please describe the amount of UBXN3B used in mock condition samples?
2. Because there is data presented from multiple viruses and cell types, it would be clearer if the authors could add additional labels to the figures. For example, add "HSV-1" in Figure 2B, 2C and 2D; "EMCV" in Figure 2F; "BMDMs" in Figure 3A; "cDCs" in Figure 3B and many more labels in Figure 4, 5 and 6.
3. In Figure 4, please label the two UBXN3B^{-/-} cells differently.

4. In Figure 4B, the authors should address why one of the UB3N3B^{-/-} cell lines exhibits impaired IFN β mRNA level upon poly I:C stimulation.

Reviewer #3 (Remarks to the Author):

In this manuscript the authors present data to support that the UB3N3B protein is a positive regulator of the STING-dependent signaling. They also claim that UB3N3B forms a complex with STING and TRIM56 to potentiate STING-dependent innate immune responses. Although it is clear that UB3N3B negatively impacts HSV-1 infection however there are several questions and concerns that the authors need to resolve before their manuscript is published. The novelty of this manuscript is the potential role of Ubx3b on the regulation of the activity STING. However, some of the data imply that Ubx3b has pleiotropic effects and some additional experiments are suggested to strengthen the manuscript. Also it is not clear whether these findings can be expanded to other systems. The authors need to explore more their findings in cells in which HSV-1 can replicate, they could explore other strains of HSV-1 since KOS is not limited passage, and they could use other pathogens that are sensed by STING to investigate whether their findings apply to other pathogens as well.

The manuscript is generally well written although on certain occasions discussed below information is missing or is not clear. Statistics support the reproducibility of the data.

Comments:

1. Fig. 1B assuming that the mock treated cells are the HEK293T-STING expressing cells why transfection with UB3N3B did not induce ISRE as in Fig. 1A? What is the amount of the UB3N3B plasmid in mock that was used for comparison with the transfected/cGAMP treated cells? Also, the amount of vector used in the cGAMP treated cells is not specified.
2. Fig. 1D same issue why UB3N3B doesn't induce ISRE if mock are the HEK293T-STING cells?
3. Fig. 2A the authors need to demonstrate that under the same conditions STING is expressed.
4. Fig. S2: The difference between Ubx3b^{+/+} and Ubx3b^{-/-} after infection is about two folds and there is Ubx3b-independent induction of innate immunity that the authors need to comment about.
5. Fig. 3D: To demonstrate the expression of STING is important.

6. Fig. 3F: growth curves of HSV-1 will make data stronger.
7. Fig. 3H: the use of β -actin mRNA to normalize the qPCR data during HSV-1 infection is not recommended as it is gradually degraded at least in cell cultures.
8. Fig.S3: ISGs are usually induced very fast upon treatment with pure IFN. ISG15 is induced by 8h after treatment with IFN. Ubxn3b is slightly upregulated only late (24 h post-treatment). It is unclear how significant is a 4-fold upregulation. Therefore, Ubxn3b expression most likely is stimulated indirectly in the presence of IFN.
9. The results in Fig. 4C and 4D are hard to interpret. Infection was done with HSV-1 at 0.5 PFU/cell and for most cells by 24 h post-infection at this PFU a lot of virus is in the supernatant and many cells have been lysed. What is the antibody used in 4D? A more detailed growth curve with more time points earlier and later, lower PFU, and a kinetic for viral gene expression are recommended. The authors will need to perform the same experiment in parallel in STING KO cells. Fig. 4C displays 3-4 log₁₀ difference between WT and Ubxn3b^{-/-} while similar studies using STING KO cells or STING KO mice have demonstrated smaller differences. The authors demonstrated that the STING KO mice are more susceptible to HSV-1 than their Ubxn3b^{-/-} mice. Fold induction of IFN-related genes is not that impressive in this experiment to explain this huge difference in virus yields and the virus is capable of blocking innate immune responses as can be seen in Fig. 4E.
10. Fig. 5A and 5B: WT virus has mechanisms to block IRF3 phosphorylation and STING activation in permissive cells. What is the role of Ubxn3b in human cells permissive to HSV-1? GAPDH protein is not recommended as loading control for HSV-1 infected cells in cell cultures.
11. Immunoprecipitation results in Fig. 5F and 5H would be stronger if only the domain responsible for the proposed interactions was missing from the full length sequence.
12. Fig. 6: There is no proof through these experiments that STING per se is ubiquitinated. The STING antibody is expected to immunoprecipitate STING and STING associated factors. Since the K150 of STING has been proposed to be ubiquitinated by TRIM56 the authors could have used that mutant of STING as a control.

Reviewer #1 (Remarks to the Author):

In this work Yang et al have examined the role of ubiquitin regulatory X domain containing proteins in the STING signaling pathway. They find that UBXN3B is essential for STING signaling and for resistance against HSV-1 infection in mice. At the mechanistic level, the authors suggest that UBXN3B interacts with STING and TRIM56, thus promoting K63-linked ubiquitination of STING and downstream signaling. The work is well designed and the data are clear. However, the mechanistic data are still somewhat underdeveloped and it is also not clear whether the phenomenon presented applies in primary human cells.

Response: We appreciate this reviewer's recognition of our study in general. We provide additional data on the mechanism of UBXN3B action as well as phenotypes in human primary cells. We show that 1) the STING phosphorylation and degradation induced by HSV-1 infection, cGAMP and ISD stimulation is significantly impaired in primary mouse and human *UBXN3B*^{-/-} compared to WT cells; 2) the immune responses to vesicular stomatitis virus (VSV) and Sendai virus (SeV), but not to encephalomyocarditis virus (EMCV) infection are modestly deficient in *UBXN3B*^{-/-} cells, a phenotype similar to *STING*^{-/-}; 3) the immune responses to a number of TLR/ RLR ligands and recombinant mouse IFN- α are similar between *Ubxn3b*^{+/+} and *Ubxn3b*^{-/-} cDCs; and 4)

MAJOR POINTS

1. *Figure 2. The murine KO cells should be stimulated through a range of PRRs and cytokine receptors and gene expression patterns should be evaluated. It is essential to characterize the specificity of UBXN3B in immune signaling*

Response: We treated primary mouse *Ubxn3b*^{+/+} and *Ubxn3b*^{-/-} DCs with several PRR ligands and recombinant mouse IFN- α . The ligands include LPS (TLR4), FSL-1 (TLR2/6), high molecular weight polyIC (TLR3/MDA5), CpG DNA (TLR9) and polyU (TLR7). The results show that UBXN3B is not essential for the TLR/RLR/IFN-JAK-STAT signaling pathways (**Supplementary Fig. 5**).

2. *Figure 3. For comparison IFN induction by a panel of RNA viruses should be tested in wt versus UBXN3B cells.*

Response: We examined the kinetics of IFN-I induction in *Ubxn3b*^{+/+} and *Ubxn3b*^{-/-} DCs by three model viruses EMCV, SeV and VSV over a time course of 24h. We noted a modest decrease in IFN-I expression stimulated by SeV and VSV at 12h after infection at the mRNA level, 12 and 24h at the protein level (**Fig. 3a-d**). However, IFN-I induction by EMCV was the same in both *Ubxn3b*^{+/+} and *Ubxn3b*^{-/-} DCs (**Fig. e,f**).

3. *The mechanistic data are somewhat superficial. As a minimum, data should be presented on phosphor-TBK1 and phospho-STING in wt versus UBXN3B cells*
4. *It is very surprising (and somewhat worrying) that none of the STING blots show the characteristic phosphor-STING band. Can the authors explain this? Along the same lines, the authors also fail to observe STING degradation following stimulation. This is also in direct contrast to most – if not all – papers on STING signaling in the literature.*

Response: We include new results on TBK phosphorylation induced by HSV-1, cGAMP and ISD in primary mouse and human cells (**Fig. 4b,d,e,f**).

We initially did not place a focus on STING phosphorylation/degradation. The failure to detect the phosphorylated form of STING is likely due to insufficient protein resolution by SDS-PAGE. Now we carefully solved this issue by using appropriate percentage (12%) of SDS-PAGE gels and extending gel running time.

Using a specific antibody against only phosphorylated human STING, we noted rapid phosphorylation of STING at 4h after cGAMP treatment. STING phosphorylation was obviously reduced and delayed in *UBXN3B*^{-/-} cells (**Fig. 4f**). In mouse primary MEFs, HSV-1 infection resulted in overt STING degradation at 12h after infection only in WT, but not in *Ubxn3b*^{-/-} cells under our experimental conditions (**Fig. 4d**). ISD treatment led to faster STING degradation in WT, and this was delayed and inhibited in *Ubxn3b*^{-/-} (**Fig. 4e**). Surprisingly STING phosphorylation was not readily observed in ISD-treated WT MEFs. We thought that this was likely because of rapid degradation. As STING is degraded at least partially via autophagy [Cell. 2013 Oct 24;155(3):688-98], we wondered if we could capture the phosphorylated STING in ISD-treated MEFs in the presence of chloroquine, a lysosomal inhibitor that can block autophagy-mediated protein degradation. Indeed, we noted phosphorylated STING in WT but not in *Ubxn3b*^{-/-} (**Fig. 4e**). These results clearly demonstrate a role for UBXN3B in STING phosphorylation. However, in WT DCs, STING was only slightly reduced (~50%) even at 24h after ISD treatment, but not at all after cGAMP or HSV-1 stimulation (**Fig. 2b**). These observations reflect cell-type specific mechanism of STING degradation and may also suggest that cDCs sustain immune responses to HSV-1 infection *in vivo*.

5. *The work would gain significantly, and could potentially compensate for the lack of detailed mechanism, of data were provided on the role of UBXN3B in primary human cells.*

Response: We recapitulate the results of mouse studies in primary human trophoblasts. The reasons to choose trophoblasts are: 1) vertical transmission of HSV-1 may involve trophoblasts, 2) like embryonic fibroblasts, trophoblasts can be passaged *in vitro* for 12-15 generations, which allows us to knockout the genes of interest with CRISPR-Cas9 and examine their functions, and 3) they can be obtained in a large quantity.

Both *IFNB1* and *TNFA* expression stimulated by cGAMP and HSV-1 was decreased significantly; while consequently HSV-1 replication was enhanced in *UNXB3B*^{-/-} (**Fig. 2h-j, 4g**).

Reviewer #2 (Remarks to the Author):

In this study, the authors generated inducible UBXN3B knockout mice and showed that the absence of UBXN3B leads to high lethality upon HSV-1 infection, which is correlated with decreased type I IFN (IFN-I) production. Moreover, HSV-1 and STING ligand-induced IFN-I was also repressed in Ubxn3b^{-/-} primary cells, indicating that UBXN3B is involved in STING dependent pathways. The authors further explored the mechanism and showed that UBXN3B interacted with both STING and one of its E3 ligases, TRIM56, to facilitate STING ubiquitination and dimerization. Most of the experiments are well done. However, there are some experimental concerns that are listed below.

Response: We appreciate this reviewer's recognition of our study in general. We provide additional data on STING ubiquitination, phosphorylation, trafficking and degradation. We show that 1) the STING phosphorylation and translocation induced by HSV-1 infection, cGAMP and ISD stimulation is partly blocked in primary mouse and human *UBXN3B*^{-/-} compared to WT cells; 2) the immune responses to vesicular stomatitis virus (VSV) and Sendai virus (SeV), but not to encephalomyocarditis virus (EMCV) infection are modestly deficient in *UBXN3B*^{-/-} cells, a phenotype similar to *STING*^{-/-}; 3) the UBXN3B-STING interaction is induced by HSV-1; 4) TBK1-STING interaction is impaired in *UBXN3B*^{-/-} cells.

1. *STING has been shown to be involved in RIG-I dependent RNA sensing pathway as well. Does RNA virus infection e.g. VSV or SeV infection have an effect on IFN responses in Ubxn3b^{-/-} mice compared to WT mice?*

Response: We performed VSV infection and noted that *Ubxn3b*^{-/-} mice were more susceptible to VSV infection than *Ubxn3b*^{+/+} (**Fig. 3a**). IFN- α induction by VSV and SeV in *Ubxn3b*^{-/-} DCs was modestly but significantly reduced compared to *Ubxn3b*^{+/+} (**Fig. 3b-d**). These results are in agreement with a previous study in *Sting*^{-/-} mice [Nature. 2008 Oct 2;455(7213):674-8; Nature. 2009 Oct 8;461(7265):788-92].

2. *Figure 4. The authors should show HSV-1 or dsDNA-induced STING ubiquitination in WT and UBXN3B^{-/-} H1975 cells. Does UBXN3B collaborate with MUL1, another E3 ligase that ubiquitinates STING?*

Response: We used primary trophoblasts that express more abundant STING than H1975 cells instead. STING ubiquitination at 3 and 6hrs after cGAMP treatment was inhibited in *UBXN3B*^{-/-} cells (**Supplementary Fig. 10**).

We attempted to use FLAG-MUL1 to pull down UBXN3B in HEK293T cells and observed no interaction between MUL1 and UBXN3B, suggesting that UBXN3B is not directly involved in MUL1 action.

MUL1 was recently shown to be critical for STING ubiquitination at the amino residue K224, consequently its translocation, phosphorylation and degradation [Sci Immunol. 2017 May 5; 2(11)]. TRIM56-mediated ubiquitination [Immunity. 2010 Nov 24;33(5):765-76] may also play a role in this process as STING K150R cannot be phosphorylated or degraded efficiently [Fig. 2E, 3G of Sci Immunol. 2017 May 5; 2(11)]. In agreement with this, UBXN3B deficiency partially blocks STING phosphorylation and trafficking (**Fig. 4 d-f**, **Supplementary Fig. 7**).

3. *UBXN3B has been shown to interact with STING under ectopic expression conditions. It would be beneficial if the authors could also show the interaction of UBXN3B and STING under endogenous conditions. Is this interaction altered upon DNA stimulation conditions, such as HSV-1 infection, cGAMP or poly I:C transfection?*

Response: **Fig.6a** shows that endogenous Sting interacted with Ubxn3b upon HSV-1 infection in MEFs.

4. The authors have shown the interaction between STING and UBXN3B. Do UBXN3B and STING also co-localize in cells? If so, does the localization of UBXN3B and STING change upon DNA stimulation?

Response: UBXN3B is known to dynamically localize to the ER, lipid droplets and mitochondria in yeast [(J Cell Sci 2012 125: 2930-2939; Mol Biol Cell. 2012 Mar;23(5):800-10]. As expected, by immunofluorescent microscopy we observed that UBXN3B localized to the perinuclear ER and partially overlapped with STING in resting cells (**Fig. 5b**). Co-localization of two proteins does not necessarily mean direct interaction, for instance Calreticulin-STING co-localization to the ER. The co-IP data demonstrated no interaction between UBXN3B and STING in resting cells (**Fig. 6a**). However, the localization pattern of UBXN3B indeed changed in part to punctate structures with STING after ISD treatment (**Fig. 5b**). These data suggest that UBXN3B regulates STING functionality at multiple locations.

5. *The authors propose that UBXN3B is an adaptor of TRIM56 and STING, therefore affecting K63-linked ubiquitination of STING by TRIM56. As TRIM56 has been shown to be critical for TBK1 recruitment to STING, is the TBK1-STING interaction affected in UBXN3B deficient cells and does this impair TBK1 phosphorylation and STING translocation?*

Response: Indeed, TBK1 phosphorylation induced by ISD/cGAMP/HSV-1 was impaired in mouse and human *UBXN3B*^{-/-} cells (**Fig. 4b, d-f**). TBK1-STING interaction (**Supplementary Fig. 10b**) and STING trafficking (**Supplementary Fig. 7**) was also disrupted.

6. *A recent publication reported that K63-linked ubiquitination is required for dsDNA-induced STING translocation, phosphorylation and degradation (Sci Immunol. 2017). Considering that UB3N3B deficiency markedly reduced K63-linked ubiquitination of STING, the authors should also check whether UB3N3B deficiency affects STING trafficking, phosphorylation and degradation in response to dsDNA stimulation.*

Response: See also the responses to Reviewer 1's Critique 3 and 4. We initially did not place a focus on STING phosphorylation/degradation. The failure to detect the phosphorylated form of STING is likely due to insufficient protein resolution by SDS-PAGE. Now we carefully solve this issue by using appropriate percentage (12%) of SDS-PAGE gels and extending gel running time. Using a specific antibody against only phosphorylated human STING, we noted rapid phosphorylation of STING at 4h after cGAMP treatment. STING phosphorylation was obviously reduced and delayed in *UB3N3B*^{-/-} cells (**Fig. 4f**). In mouse primary MEFs, HSV-1 infection resulted in overt STING degradation at 12h after infection only in WT, but not in *Ubx3b*^{-/-} cells under our experimental conditions (**Fig. 4d**). ISD treatment led to faster STING degradation in WT, and this was delayed and inhibited in *Ubx3b*^{-/-} (**Fig. 4e**). Surprisingly STING phosphorylation was not readily observed in ISD-treated WT MEFs. We thought that this was likely because of rapid degradation. As STING is degraded at least partially via autophagy [Cell. 2013 Oct 24;155(3):688-98], we wondered if we could capture the phosphorylated STING in ISD-treated MEFs in the presence of chloroquine, a lysosomal inhibitor that can block autophagy-mediated protein degradation. Indeed, we noted phosphorylated STING in WT but not in *Ubx3b*^{-/-} (**Fig. 4e**). These results clearly demonstrate a role for UB3N3B in STING trafficking, phosphorylation and degradation.

7. *All figure legends need to indicate how many times the experiment was done and statistical significance needs to be shown for all experiments.*

Response: Per suggestions, we include the information in the revision.

Minor:

1. *Figure 1B and 1C. Please describe the amount of UB3N3B used in mock condition samples?*

Response: The information is indicated in the figures (**Supplementary Fig. 1b,c**).

2. *Because there is data presented from multiple viruses and cell types, it would be clearer if the authors could add additional labels to the figures. For example, add "HSV-1" in Figure 2B, 2C and 2D; "EMCV" in Figure 2F; "BMDMs" in Figure 3A; "cDCs" in Figure 3B and many more labels in Figure 4, 5 and 6.*

Response: The information is indicated in the figures (**Fig. 2, Fig. 3, and Supplementary Fig. 5**).

3. *In Figure 4, please label the two UB3N3B^{-/-} cells differently.*
4. *In Figure 4B, the authors should address why one of the UB3N3B^{-/-} cell lines exhibits impaired IFN β mRNA level upon poly I:C stimulation.*

Response: As the decrease is modest, we believe that this is likely a non-specific off-targeting effect of this guide RNA. To verify if UB3N3B has a role in polyIC-stimulated immune signaling, we performed polyIC treatment in primary mouse cDCs. The results show no difference in IFN-I expression between *Ubx3b*^{+/+} and *Ubx3b*^{-/-} cells (**Supplementary Fig. 5d**). In light of the new data and in order not to mislead readers, we removed this clone out of the figures (**Supplementary Fig. 4b**).

Reviewer #3 (Remarks to the Author):

In this manuscript the authors present data to support that the UBXLN3B protein is a positive regulator of the STING-dependent signaling. They also claim that UBXLN3B forms a complex with STING and TRIM56 to potentiate STING-dependent innate immune responses. Although it is clear that UBXLN3B negatively impacts HSV-1 infection however there are several questions and concerns that the authors need to resolve before their manuscript is published. The novelty of this manuscript is the potential role of Ubxn3b on the regulation of the activity STING. However, some of the data imply that Ubxn3b has pleiotropic effects and some additional experiments are suggested to strengthen the manuscript. Also it is not clear whether these findings can be expanded to other systems. The authors need to explore more their findings in cells in which HSV-1 can replicate, they could explore other strains of HSV-1 since KOS is not limited passage, and they could use other pathogens that are sensed by STING to investigate whether their findings apply to other pathogens as well.

The manuscript is generally well written although on certain occasions discussed below information is missing or is not clear. Statistics support the reproducibility of the data.

Response: We appreciate this reviewer's recognition of our study in general. In the revision we provide additional data on the mechanism of UBXLN3B action, phenotypes with HSV-1 permissive primary MEFs and human cells, infection with several RNA viruses that can activate STING signaling and stimulation with a panel of TLR/PRR ligands (see also Responses to Reviewer 1 and 2).

Comments:

1. *Fig. 1B assuming that the mock treated cells are the HEK293T-STING expressing cells why transfection with UBXLN3B did not induce ISRE as in Fig. 1A? What is the amount of the UBXLN3B plasmid in mock that was used for comparison with the transfected/cGAMP treated cells? Also, the amount of vector used in the cGAMP treated cells is not specified.*
2. *Fig. 1D same issue why UBXLN3B doesn't induce ISRE if mock are the HEK293T-STING cells?*

Response: HEK293 cells express barely detectable cGAS and STING. To study STING signaling in HEK293 cells, we thus employed two approaches: transient overexpression (**Supplementary Fig. 1A**) and reconstituted stable expression (**Supplementary Fig. 1B**). In Supplementary Fig. 1A, STING was transiently overexpressed together with UBXLNs and the induction of ISRE was relative to empty plasmid control (no STING, no UBXLN, the first Vec). Transient overexpression of STING alone stimulated ISRE by ~150 fold (the second Vec+ HA-STING). ISRE was further enhanced by ~8 fold in the presence of both STING and UBXLN3B. In Supplementary Fig. 1B, a **stable STING-expressing HEK293T** cell line was employed. In this scenario, activation of STING signaling requires an agonist such as cGAMP (similar results with human STING-reconstituted MEFs [Fig 2A, E, Sci Immunol. 2017 May 5;2(11), pii: eaah7119]. Transient overexpression of UBXLN3B alone without cGAMP (the second Mock bar) failed to induce ISRE when compared to Vec (the first in Mock). Under physiological conditions, TRIM56 may need cellular cues to be activated first before ubiquitinating STING. UBXLN3B may serve as an adaptor to facilitate activated TRIM56 and STING interaction, rather than a direct activator of TRIM56.

The plasmid amount is indicated in the figures now.

3. *Fig. 2A the authors need to demonstrate that under the same conditions STING is expressed.*

Response: We noted high STING expression in the lung and spleen, but very low or undetectable in other tissues tested (brain, liver, kidney, heart) (**Fig. 1a**). This result is consistent with human STING expression pattern (<https://www.proteinatlas.org/ENSG00000184584-TMEM173/tissue>).

4. *Fig. S2: The difference between $Ubxn3b^{+/+}$ and $Ubxn3b^{-/-}$ after infection is about two folds and there is $Ubxn3b$ -independent induction of innate immunity that the authors need to comment about.*

Response: We agree. In leukocytes, other dsDNA sensing pathways may be also important for the immune responses to HSV-1 infection. For example in pDCs TLR9 senses non-methylated CpG DNA and HSV-1 DNA has been shown to efficiently stimulate TLR9 responses [Nucleic Acids Res. 2008 May; 36(9): 2825–2837; Blood. 2004;103:1433–1437]. In non-TLR9 expressing, STING-expressing cells such as lung epithelial cells, and etc. the STING signaling may be dominant in mounting immune responses to HSV-1. In addition, other E3 ligases such as AMFR [Immunity. 2014 Dec 18;41(6):919-33], MUL1 [Sci Immunol. 2017 May 5; 2(11): eaah7119] also contribute to STING activation. We added these to the discussion (page 13, lines 311-313).

5. *Fig. 3D: To demonstrate the expression of STING is important.*

Response: See also the response to Rev 1 Q3/4. STING protein expression is now included (**Fig. 2b**).

6. *Fig. 3F: growth curves of HSV-1 will make data stronger.*

Response: We include 6, 12, 24, 36 and 48h time points which cover all the stages of HSV-1 infection (**Fig. 2f, g**).

7. *Fig. 3H: the use of β -actin mRNA to normalize the qPCR data during HSV-1 infection is not recommended as it is gradually degraded at least in cell cultures.*

Response: We changed to hydroxymethylbilane synthase (HMBS) (**Fig. 2g**). A previous study demonstrated that HMBS was the least unaffected housekeeping gene during HSV-1 infection; PPIA was the most stable one in other viral infections [Virology Journal 20074:130].

8. *Fig.S3: ISGs are usually induced very fast upon treatment with pure IFN. ISG15 is induced by 8h after treatment with IFN. $Ubxn3b$ is slightly upregulated only late (24 h post-treatment). It is unclear how significant is a 4-fold upregulation. Therefore, $Ubxn3b$ expression most likely is stimulated indirectly in the presence of IFN.*

Response: Based on the fold induction (~6 fold) of Isg15, IFN- α stimulation is relatively mild in this experiment (**Supplementary Fig. 6b**). We repeated IFN- α treatment in cDCs and observed that $Ubxn3b$ was gradually and constantly up-regulated throughout IFN- α treatment, though not as drastically as Isg15 (**Supplementary Fig. 6a**).

9. The results in Fig. 4C and 4D are hard to interpret. Infection was done with HSV-1 at 0.5 PFU/cell and for most cells by 24 h post-infection at this PFU a lot of virus is in the supernatant and many cells have been lysed. What is the antibody used in 4D? A more detailed growth curve with more time points earlier and later, lower PFU, and a kinetic for viral gene expression are recommended. The authors will need to perform the same experiment in parallel in STING KO cells. Fig. 4C displays 3-4 log₁₀ difference between WT and $Ubxn3b^{-/-}$ while similar studies using STING KO cells or STING KO mice have demonstrated smaller differences. The authors demonstrated that the STING KO mice are more susceptible to HSV-1 than their $Ubcn3b^{-/-}$ mice. Fold induction of IFN-related genes is not that impressive in this experiment to explain this huge difference in virus yields and the virus is capable of blocking innate immune responses as can be seen in Fig. 4E.

Response: First we need to clarify that the scale is not log scale in Fig. 4C of the first submission (**now Supplementary Fig. 4C**), and thus the difference in HSV-1 titers is actually ~7-8 fold at 24h. Nonetheless we

repeated the experiments including more time points in primary human trophoblasts in parallel with *STING*^{-/-}. In our experimental conditions (virus passage, cell types and etc), the cells were healthy before 36 hrs infection. By fluorescent microscopy, we noted similar viral loads (HSV-GFP) in *UBXN3B*^{-/-} and *STING*^{-/-}, both were higher than WT (**Fig. 2i**).

The anti-HSV-1/HSV-2 antiserum is from Antibodies-online (Cat# ABIN285517). It is specific for the ICP's and late structural (virion) antigens (<https://www.antibodies-online.com/antibody/285517/anti-Herpes+Simplex+Virus+1/2+HSV1/HSV2+antibody/>).

9. *Fig. 5A and 5B: WT virus has mechanisms to block IRF3 phosphorylation and STING activation in permissive cells. What is the role of Ubxn3b in human cells permissive to HSV-1? GAPDH protein is not recommended as loading control for HSV-1 infected cells in cell cultures.*

Response: We have also included tubulin. In the early stage of infection (8hrs), GAPDH /actin expression did not change significantly when compared to tubulin (**Fig. 4a,b**).

We observed similar consistent results in human primary trophoblasts as MEFs (**Fig. 2h-j**) and H1975 cell line (**Supplementary Fig. 4c-e**).

10. *Immunoprecipitation results in Fig. 5F and 5H would be stronger if only the domain responsible for the proposed interactions was missing from the full length sequence.*

Response: The Δ UAS and Δ CC mutant of *UBXN3B* did not bind *STING* or *TRIM56* respectively (**Supplementary Fig. 8**).

11. *Fig. 6: There is no proof through these experiments that STING per se is ubiquitinated. The STING antibody is expected to immunoprecipitate STING and STING associated factors. Since the K150 of STING has been proposed to be ubiquitinated by TRIM56 the authors could have used that mutant of STING as a control.*

Response: We repeated the experiment including K150R (**Supplementary Fig. 10a**).

Reviewers' comments:

Reviewer #1 (Remarks to the Author):

This reviewer finds that the authors have complied with the points raised in a satisfactory manner, and that the work is now improved significantly.

Reviewer #2 (Remarks to the Author):

The authors have adequately addressed my previous concerns.

Reviewer #3 (Remarks to the Author):

The authors have made a considerable effort to improve the manuscript but a few points are still unclear:

1. If in Fig. 2F the infection was done at 0.1 PFU/cell then it is difficult to reconcile that at 12 h post infection this level of replication is recorded. A 2-3 h time point is missing to compare the input virus.
2. The degradation of STING via autophagy in MEF infected cells figure 4d seems an over interpretation. At 12h p.i. the levels of STING in Ubcxn3b^{+/+} are similar to the uninfected cells at time 0. In human cells STING is not degraded during HSV-1 infection. More convincing data are required.
3. The quality of p-TBK1 in Fig. 4f is poor creating problems to interpret.
4. In Fig. 5b ISD the punctate structures are not convincing. The Ubcxn3b little punctate on the left side of the nucleus do not actually colocalize with STING. STING in this area displays a diffuse pattern. Better quality picture may help.
5. The suppl. Fig. 10a didn't provide any clarity.

Reviewer #3 (Remarks to the Author):

The authors have made a considerable effort to improve the manuscript but a few points are still unclear:

Response: We thank Reviewer 3 for his/her favorable and constructive comments on the manuscript. We have repeated and performed additional experiments to address all the concerns.

1. If in Fig. 2F the infection was done at 0.1 PFU/cell then it is difficult to reconcile that at 12 h post infection this level of replication is recorded. A 2-3 h time point is missing to compare the input virus.

Response: This is a valid point and we have repeated these experiments including 2, 12, 24 36 and 48h time points. We incubated MEFs with HSV-1 for 2 hours, removed inoculum, washed with 1XPBS once and then replaced with fresh medium. We measured extracellular HSV-1 particles by plaque forming assay at 2, 12, 24 36 and 48h after PBS wash. HSV-1 titers were low at 2h and increased to $\sim 10^4$ PFU/ml in WT cells at 12h. The titers in knockout cells were consistently higher than WT at 12, 24 and 36h.

In addition, we think that the HSV-1 replication kinetics and titers produced by MEFs could vary with HSV-1 infectivity and sources of MEFs. Particularly, MEF permissiveness to HSV-1 may vary with mouse background. For example, C57BL/6 MEFs produced 1×10^4 viruses per million cells (in 3ml medium) by 12hr after infection (MOI=5), and 2×10^5 PFU by 24hr; while BALB/c MEFs produced 2×10^6 viruses by 24hr (MOI=5) (Fig. 2, *Invest Ophthalmol Vis Sci* 27:909-914, 1986). A MEF line (129/SV x C57BL/6) produced 10^7 /ml PFU by 24hr after infection (MOI=0.1) [Fig. 4, *Proc Natl Acad Sci U S A.* 101(6): 1473–1478, 2004], and 129SV/EV MEFs made $\sim 8 \times 10^6$ PFU/ml at 24hr infection (MOI=0.01) [Fig. 1B, *Virology*, 450–451: 350-354, 2014].

The detailed method we used to titrate the extracellular HSV-1 is as following:

Day1: we seeded sub-confluent MEFs to 12 well plates.

Day3: we collected one well cell and counted the cell number. Based on the cell number we counted to set up the titer for infection. In this case, we used 0.1 MOI HSV-1 to infect MEFs. We diluted HSV-1 with serum free DMEM medium and then infected the MEFs for 2 hours followed by washing with PBS for once. After washing, we added complete DMEM culture medium and set it as 0 hour. We collected the supernatant at 0, 2, 12.. hours, respectively.

Day4: we seeded 0.5×10^6 Vero cells in 6-well plates.

Day5: we diluted the supernatant collected at different time point with serum free DMEM and then infected the Vero cells for 2 hours. After infection, we removed the supernatant followed by adding 2ml 1% low-melting agarose dissolved in complete DMED culture medium.

Day8: we counted the plaques after staining with Neutral Red.

2. The degradation of STING via autophagy in MEF infected cells figure 4d seems an over interpretation. At 12h p.i. the levels of STING in Ubxn3b^{+/+} are similar to the uninfected cells at time 0. In human cells STING is not degraded during HSV-1 infection. More convincing data are required.

Response: We would like to point out that the visual difference in Sting level between 0 and 12hr Ubxn3b^{+/+} is not striking due to unequal loading. Sting is actually degraded in Ubxn3b^{+/+} MEFs at 12h if normalized with loading control-Tubulin. We repeated the experiment and resolved Sting much better. We noted Sting phosphorylation at 6h after infection and obvious degradation by 12h after infection in Ubxn3b^{+/+} but not in Ubxn3^{-/-} MEFs (Fig. 4d). We also performed the same experiment in human cells and similar results were observed (Supplementary Fig.7).

3. The quality of p-TBK1 in Fig. 4f is poor creating problems to interpret.

Response: We have replaced it with an improved blot.

4. In Fig. 5b ISD the punctate structures are not convincing. The Ubcxn3b little punctate on the left side of the nucleus do not actually colocalize with STING. STING in this area displays a diffuse pattern. Better quality picture may help.

Response: We have replaced it with an improved image.

5. The suppl. Fig. 10a didn't provide any clarity.

Response: We repeated the experiment and provide new blots in the revision.

REVIEWERS' COMMENTS:

Reviewer #3 (Remarks to the Author):

Questions have been addressed. Thanks.